# Ice/firn age distribution on the Elbrus Western Plateau (Caucasus) inferred from ice flow model

Gleb Chernyakov[1], Nelly Elagina[1], Taisiia Kiseleva[1], and Stanislav Kutuzov[2, 3]

[1]Department of Glaciology, Institute of Geography, Russian Academy of Sciences, Moscow, Russia
[2]Byrd Polar and Climate Research Center, The Ohio State University, Columbus, OH, USA
[3]School of Earth Sciences, The Ohio State University, Columbus, OH, USA

**Correspondence:** Gleb Chernyakov (glchern@igras.ru)

**Abstract.** The glaciers of Mount Elbrus (Caucasus) contain paleoclimatic and paleoenvironmental information representative of a vast region. Negligible seasonal melting in the near-summit area of Elbrus ensures excellent preservation of climatic signals. In 2009, a 182.65 meter long ice core was obtained from the glacier on the near-summit Western Plateau (WP) of Elbrus. The upper part and basal samples of the core were dated. In this work, a three-dimensional (3D) steady state thermomechanically coupled Stokes flow model for a cold glacier with a rheological law accounting for firn densification, calibrated based on the ice core dating, was applied to model the velocity field and the corresponding distribution of the age of the ice in the central part of the WP. We performed multiple model runs, varying boundary conditions (BCs), ice viscosity, and the inclusion of thermomechanical coupling. The Elmer/Ice software was used for numerical simulation. The model quite accurately reproduces the age of the ice according to ice core data to a depth of 165 m (up to 243 years). Below, the age of the ice increases sharply and the discrepancies in dating between different modeling scenarios become larger. Overall, the simulated ages fell within 68.2 % confidence intervals for the ages of near-bottom ice samples (mean radiocarbon age 1–2 ka). The model is not applicable for dating the lowermost ice layer (3–4 m thick). Future model improvements should focus on accounting for potential melting and identifying areas containing the oldest ice.

## 1   Introduction

The glaciers of Mt. Elbrus offer a unique paleoclimate archive that traces signals from a large region, including the North Caucasus, the Black Sea region, Southeastern Europe, North Africa, and the Middle East (Mikhalenko et al., 2024; Kutuzov et al., 2019a). The cold conditions (10 m depth temperature of $-17.3\,^\circ$C) and negligible seasonal melting in the near-summit region of Elbrus ensure the preservation of climatic and environmental signals in ice cores (Mikhalenko et al., 2015).

The Elbrus Western Plateau (WP) has been the subject of ice core studies since 2004 (Mikhalenko et al., 2005). Glaciochemical investigations of a deep ice core (182.65 m long – from the glacier surface to solid rock) drilled in 2009 include reconstructions of past anthropogenic sulfur emissions (Preunkert et al., 2019), black carbon (Lim et al., 2017), dust events (Kutuzov et

al., 2013, 2019a), ammonia (Legrand et al., 2025), dissolved organic carbon (Legrand et al., 2024), polycyclic aromatic hydrocarbons (Vecchiato et al., 2020), isotopic composition (Kozachek et al., 2017), and snow accumulation (Mikhalenko et al., 2024).

Relatively high annual accumulation (1.2 m w.e.) ensures the preservation of the seasonal cycle in geochemistry down to a depth of 168.6 m (131.6 m w.e.) in the ice core. This section was dated by annual layer counting, primarily using pronounced seasonal variations in ammonium and succinate concentrations (Preunkert et al., 2019). According to an estimate based on a two-dimensional (2D) analytical model of Salamatin et al. (2000), the age of the ice at the bottom of the drilling site does not exceed 350–400 years and the age of the basal ice in the deepest part of the glacier (more than 250 m deep) is about 660 years (Mikhalenko et al., 2015). Radiocarbon dating indicates that bottom ice samples from a depth of 176.89–182.15 m are between 1 and 2 ka old (Preunkert et al., 2019). Detailed ground-penetrating radar (GPR) surveys (Lavrentiev et al., 2010; Kutuzov et al., 2019b) revealed that the glacier ice at the WP site fills an ancient volcanic crater, resulting in specific morphological conditions (a large aspect ratio) and thermodynamic conditions (a high geothermal heat flux). More recently, the ice flow modeling was performed for the WP using a three-dimensional (3D) Stokes ice flow model using only dynamical equations to account for the upstream effect in the accumulation record (Mikhalenko et al., 2024). Despite significant progress in glaciological investigations of this site, several important questions remain unresolved, including the spatial distribution of the age of the ice.

The objectives of this study are to determine the spatial distribution of the ice/firn age at the WP and, in particular, to reconstruct the age of the intermediate section of the 2009 core, which had not previously been dated by other methods. We apply a thermomechanically coupled 3D flow model of the present-day glacier, based on the finite element modeling software Elmer/Ice (Gagliardini et al., 2013), which solves 3D Stokes equations for a nonlinear viscous fluid with the firn rheological law of Gagliardini and Meyssonnier (1997).

Ice flow models have been repeatedly used to study the age distribution of ice in mountain glaciers and, in particular, ice cores. Dating models have undergone a natural evolution - from 2D analytical solutions to 3D numerical approaches that take into account the mutual influence of mechanical and thermal processes. The main stages of this development appear to us as follows. 2D purely mechanical (Vincent et al., 1997) and thermomechanically coupled (Salamatin et al., 2000; Shiraiwa et al., 2001) analytical models were developed and applied for modeling age–depth relations in ice at Dôme du Goûter (Mont Blanc, French Alps) and at Ushkovsky Volcano (Kamchatka Peninsula), respectively. Gagliardini and Meyssonnier (1997) adapted the rheological law of Duva and Crow (1994) for a cold glacier with a thick firn layer and implemented it in a 2D dynamical finite element model for Dôme du Goûter. Further, the firn rheological law of Gagliardini and Meyssonnier (1997) was applied in 2D and 3D finite element models for Colle Gnifetti glacier saddle (Monte Rosa, Swiss/Italian Alps) in the work of Lüthi and Funk (2000). Zwinger et al. (2007) modified the rheological relations of Gagliardini and Meyssonnier (1997) for 3D thermomechanically coupled Stokes flow model implemented based on Elmer/Ice and applied the model to a crater glacier at Ushkovsky Volcano. Konrad et al. (2013) used a semi-analytical 2D flow model with Glen's rheology in combination with the data of ice core and GPR for determining the age distribution of Colle Gnifetti. Gilbert et al. (2014) performed a simulation of firn age for the Col du Dôme glacier (Mont Blanc area) to validate their 3D thermomechanically coupled transient model,

including the firn densification process, developed for modeling the thermal regime of polythermal mountain glaciers and implemented in Elmer/Ice. Liciulli et al. (2020) applied a 3D Stokes thermomechanically coupled model with firn rheology implemented in Elmer/Ice to interpret ice core data at Colle Gnifetti and included this model to a more general transient simulation scheme.

## 2  Study area

The glaciated area of Mt. Elbrus is around 112 $km^2$. Glaciers cover an altitudinal range from 2680 to 5642 m a.s.l. Above 5200 m a.s.l. temperature stays negative throughout a year and no melting occurs (Mikhalenko et al., 2020). Elbrus glaciers are currently losing mass ($-0.55 \pm 0.04$ m w.e. $a^{-1}$). The rate of Elbrus glacier mass loss tripled in 1997–2017 compared with the 1957–1997 period (Kutuzov et al., 2019b). The ice cores used for climate and environment reconstructions were recovered at the WP at 5115 m a.s.l. (Mikhalenko et al., 2020). A 182.65 m ice core was recovered at the WP at a point with coordinates $43°20'53.9''$ N and $42°25'36.0''$ E in August–September 2009. The WP which covers approximately 0.5 $km^2$, is bordered by lava ridges to the south and southeast, and a vertical wall of Mt. Elbrus to the east (Fig. 1). The results of a series of ground-based radar surveys at a frequency of 20 MHz in 2005, 2007, and 2017 show a significant ice thickness and a crater shape of the underlying bedrock. The maximum depth is $255 \pm 8$ m at the central part of the plateau, with minimum values of about 60 m near the edge. The GPR survey used in this study was conducted in July 2017. The ice thickness map was completed using empirical Bayesian kriging interpolation (Kutuzov et al., 2019b). The 10 m depth temperature is $-17.3\,°C$ (Mikhalenko et al., 2015). A series of field measurement campaigns were conducted on the WP in 2004–2018, including the detailed low and high frequency GPR surveys, snow accumulation distribution measurements, meteorological measurements, snow pits and ice cores from several shallow and three deep (182.65, 150.3, and 119.8 m) boreholes (Mikhalenko et al., 2020). For the information about topography, we used the Pléiades digital elevation model (DEM) from 8 September 2017 with the vertical uncertainty between $\pm0.5$ m and $\pm1$ m (Kutuzov et al., 2019b). The map of the subglacial relief of the WP was constructed as the difference between the elevation marks of the glacier surface according to the DEM and the data on the ice thickness (Mikhalenko et al., 2020).

## 3  Research data

To model the flow and determine the age of ice/firn in the WP glacier, we used the following data:

1. DEMs of the glacier surface and bedrock with a cell size of $10 \times 10$ m, both obtained in 2017.

2. Temperature distribution in a deep-drilling borehole obtained from measurements in 2009.

3. Ice/firn density distribution in the 2009 deep ice core.

4. Ice/firn age distribution in the 2009 ice core with a resolution of about one year, obtained by laboratory methods (upper 168.6 m).

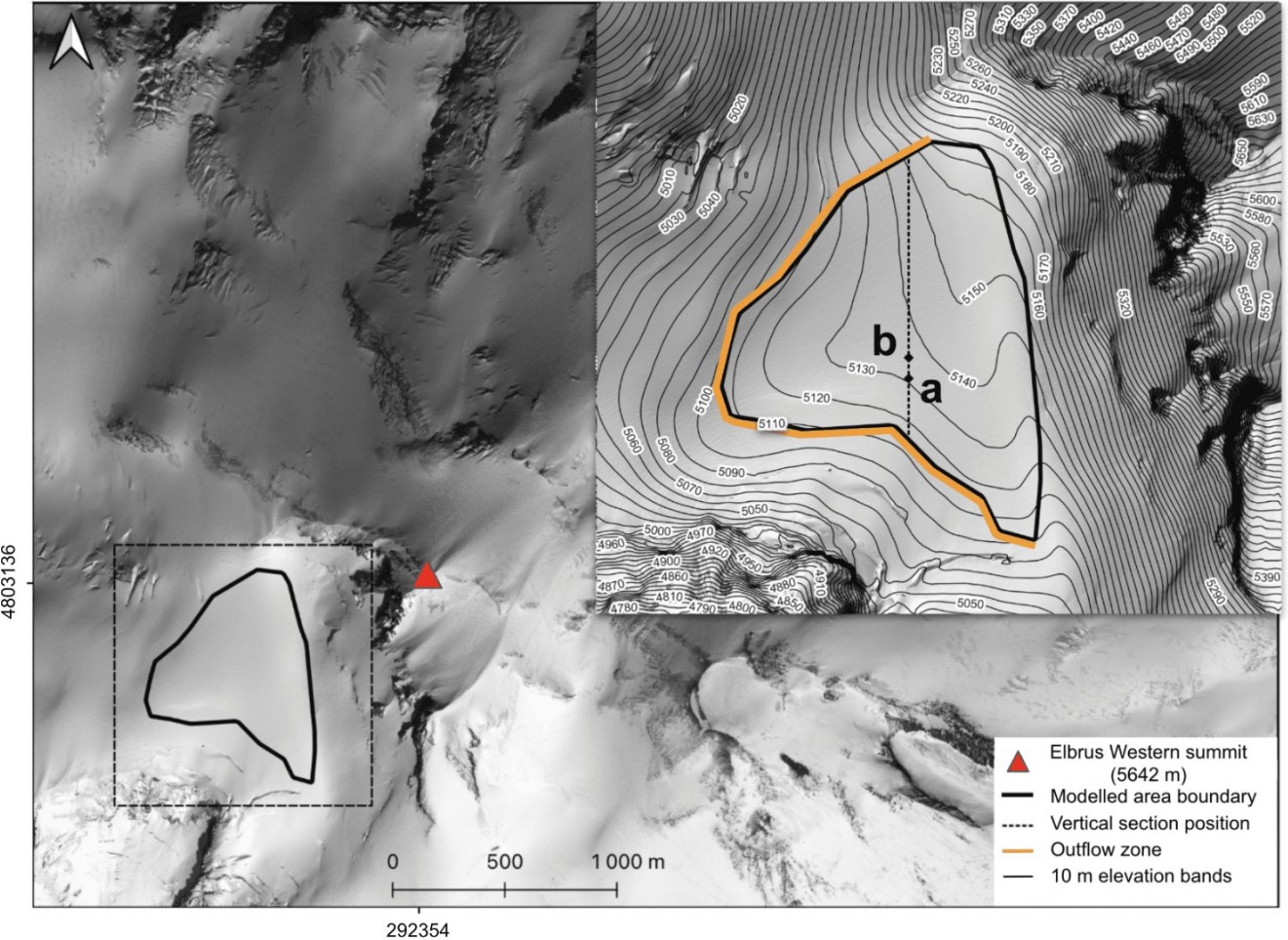

**Figure 1.** Study area on Mt. Elbrus. The vertical distributions obtained in our study (Fig. 5) refer to location $a$; the 2009 drilling site is marked with $b$. The top-right figure is a zoom of the black dashed square. The elevation bands are based on the Pléiades DEM of 2017. The coordinates are presented in the WGS 84 (UTM 38 N). The SPOT 7 image obtained on 20 August 2016 is shown as a background.

5. Interval age estimates of basal ice samples obtained by the radiocarbon method (lower 5.8 m).

## 4  Spatial structure

Ice/firn flow modeling with subsequent dating was performed in a 3D domain (Fig. 2). The domain is limited by that part of the glacier on the WP, for which DEMs of both the surface and the bed are available. The computational domain is bounded by three surfaces: a part of the glacier surface, the lateral surface of the domain (the vertical "wall"), and a part of the glacier base.

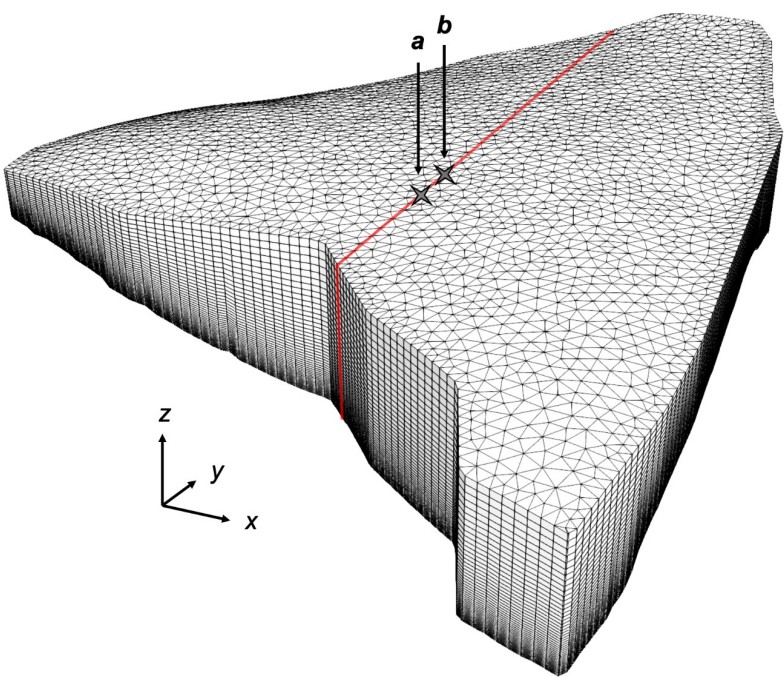

**Figure 2.** Computational domain and computational grid. The vertical distributions obtained in our study (Fig. 5) refer to location $a$; the 2009 drilling site is marked with $b$. The vertical section to which the simulation results presented in Fig. 4 relate is shown by the red line. Directions of coordinate axes: $x$ – eastward, $y$ – northward, $z$ – upward.

We chose to evaluate the model results at position $a$ instead of position $b$, where the borehole was located (see Fig. 2), for the following reason. According to the DEMs of the surface and bedrock, obtained by interpolation of the GPR data profiles the ice thickness at the 2009 drilling site location is 6.1 m greater than the ice core length. This difference is within the accuracy of the GPR data but creates the discrepancy in modeled ice thickness. Possible reasons for this discrepancy include: time difference in obtaining the DEMs of the surface and the base (8 years); errors in constructing the DEMs; inaccuracy in determining the coordinates of the drilling site; probable presence of a solid inclusion in the ice above the glacier bed, which could become a mechanical obstacle to drilling to the bed (Stiévenard et al., 1996).

In our opinion, comparison of age–depth distributions (one modeled and one ice core based) for locations at a glacier with different ice thicknesses makes little sense. Also, the glacier surface elevation at the drilling site (5135.42 m a.s.l.) obtained using the 2017 DEM does not align closely with the glacier surface elevation at the drilling site determined during deep drilling in 2009 (5115 m a.s.l.; Mikhalenko et al. (2015)), probably due to an assumed error in determining the latter. This makes problematic another possible interpretation of the original data, namely, matching the zero depth of the ice core with the surface altitude of the computational domain.

To represent the vertical profiles of ice/firn age and temperature and compare them with ice core data, we chose one of the points closest to the drilling site with similar ice thickness (182.67 m according to DEMs) and topography of surface and bedrock. This point is located 50 m south of the drilling site (denoted by *a* in Fig. 1, 2, and 4).

## 5   Ice flow model

Under the assumption of steady state, the velocity distribution in the glacier allows one to calculate the time required for each ice/firn particle to move from the glacier surface to its current position. The calculation of the velocity field was performed on the basis of a 3D stationary Stokes model with the rheological law of Gagliardini and Meyssonnier (1997) for a compressible nonlinear viscous medium (ice/firn). The model is applied in its full version, without scaling and excluding any terms in the equations below. We have applied both purely mechanical (isothermal) and thermomechanically coupled ice flow models.

Information on the quantities used in the models is given in Table 1.

Table 1: Quantities. This table describes most of the quantities used in the ice flow model. The units in the table are chosen to be more representative, rather than coherent.

| Group | Quantity | Description | Value / Dependent on | Units |
|---|---|---|---|---|
| Geometric | $x, y$ | Horizontal coordinates | Independent variables | m |
| | $z$ | Upward vertical coordinate | Independent variable | m |
| | $s$ | Glacier surface altitude | $x, y$ | m |
| | $d$ | Depth | $x, y, z$ | m |
| | $H$ | Ice thickness | $x, y$ | m |
| | $\boldsymbol{n}$ | Outer unit normal vector to the computational domain boundary | $x, y, z$ | Dimensionless |
| Mechanical | $\rho$ | Ice/firn density | $d$ | $\mathrm{kg\,m^{-3}}$ |
| | $\rho_\mathrm{i}$ | Pure ice density | 918 | $\mathrm{kg\,m^{-3}}$ |
| | $\varphi$ | Relative density | $\rho$ | Dimensionless |
| | $p$ | Pressure | Free field | Pa |
| | $\boldsymbol{\sigma}$ | Cauchy stress tensor | $p, \mu, \mathbf{D}$ | Pa |
| | $n$ | Stress exponent | 3 | Dimensionless |
| | $\boldsymbol{v}$ | Velocity | Free field | $\mathrm{m\,a^{-1}}$ |
| | $v_\mathrm{b}$ | Basal normal velocity | $10^{-6}$ | $\mathrm{m\,a^{-1}}$ |
| | $v_\mathrm{out}$ | Lateral outflow velocity | $d$ | $\mathrm{m\,a^{-1}}$ |
| | $v_\mathrm{max}$ | Maximum lateral outflow velocity | 20 | $\mathrm{m\,a^{-1}}$ |
| | $\mathbf{D}$ | Strain-rate tensor | $\mathrm{grad}\,\boldsymbol{v}$ | $\mathrm{a^{-1}}$ |
| | $\boldsymbol{g}$ | Gravitational acceleration | 9.81 | $\mathrm{m\,s^{-2}}$ |
| | $\mu$ | Shear viscosity | $T', \delta, \varphi$ | $\mathrm{Pa\,s}$ |
| | $E$ | Flow enhancement factor | 0.016, 0.2, 0.25 | Dimensionless |
| Thermodynamic | $T$ | Temperature | Free field | K |
| | $T_\mathrm{m}$ | Melting temperature of ice | $p$ | K |
| | $T_0$ | Melting temperature of ice for low pressure | 273.15 | K |
| | $T'$ | Temperature relative to the pressure melting point | $p, T$ | K |
| | $T_\mathrm{s}$ | Glacier surface temperature | $-18$ | °C |
| | $\beta$ | Clausius-Clapeyron constant for air-saturated ice | $9.8 \times 10^{-8}$ | $\mathrm{K\,Pa^{-1}}$ |
| | $A$ | Rate factor | $2.291 \times 10^{-25}$ / $T'$ | $\mathrm{Pa^{-3}\,s^{-1}}$ |

| | | | |
|---|---|---|---|
| $A_0$ | Pre-exponential constant | $\begin{cases} 3.985 \times 10^{-13} & (T' \le 263.15 \text{ K}) \\ 1.916 \times 10^3 & (T' > 263.15 \text{ K}) \end{cases}$ | $\text{Pa}^{-3}\,\text{s}^{-1}$ |
| $Q$ | Activation energy | $\begin{cases} 60 & (T' \le 263.15 \text{ K}) \\ 139 & (T' > 263.15 \text{ K}) \end{cases}$ | $\text{kJ}\,\text{mol}^{-1}$ |
| $R$ | Universal gas constant | 8.314 | $\text{J}\,\text{mol}^{-1}\,\text{K}^{-1}$ |
| $c$ | Specific heat | $T$ | $\text{J}\,\text{kg}^{-1}\,\text{K}^{-1}$ |
| $k$ | Heat conductivity | $T$ | $\text{W}\,\text{m}^{-1}\,\text{K}^{-1}$ |
| $\boldsymbol{q}$ | Heat flux | $T, \operatorname{grad} T$ | $\text{W}\,\text{m}^{-2}$ |
| $q_{\text{geo}}$ | Geothermal flux | 0.34 | $\text{W}\,\text{m}^{-2}$ |
| Other | $t$ | Time | Independent variable | a |
| | $\mathcal{A}$ | Ice/firn age | Free field | a |
| | $a, b$ | Flow law auxiliary functions | $\varphi$ | Dimensionless |
| | $\delta$ | Tensor invariant | $\mathbf{D}, \varphi$ | $\text{s}^{-1}$ |
| | $\mathbf{I}$ | Unit tensor | $\operatorname{diag}(1,1,1)$ | Dimensionless |

## 5.1 Dating problem

In order to obtain a 3D age field of ice/firn $\mathcal{A}(x,y,z)$, we first calculated the velocity field $\boldsymbol{v}(x,y,z)$ in the computational domain and then solved the dating equation $\mathrm{d}\mathcal{A}/\mathrm{d}t = 1$, or

$$\frac{\partial \mathcal{A}}{\partial t} + \boldsymbol{v} \cdot \operatorname{grad} \mathcal{A} = 1 \tag{1}$$

with a boundary condition of zero age at the surface of the glacier:

$$\mathcal{A}|_{\text{s}} = 0. \tag{2}$$

Due to the steady-state assumption $\partial \mathcal{A}/\partial t \equiv 0$.

## 5.2 Constitutive relations

The velocity field calculation for the WP glacier was performed based on a 3D steady-state Stokes flow model with the
rheological law of Gagliardini and Meyssonnier (1997) for a compressible nonlinear viscous medium (ice/firn). The mathematical formulation of the ice/firn flow problem follows the work of Zwinger et al. (2007) with some modifications.

Based on the surface heights, a depth field is calculated in the entire 3D domain:

$$d(x,y,z) = s(x,y) - z. \tag{3}$$

Based on an approximation of the density distribution in the 2009 ice core, the ice/firn density in the glacier is represented as a function of depth (Fig. 3) as follows :

$$\rho(d) = \begin{cases} \rho_i\left(1 - 0.56e^{-0.028d}\right), & d < 104.5 \text{ m} \\ 0.97\rho_i, & d \geq 104.5 \text{ m}. \end{cases} \tag{4}$$

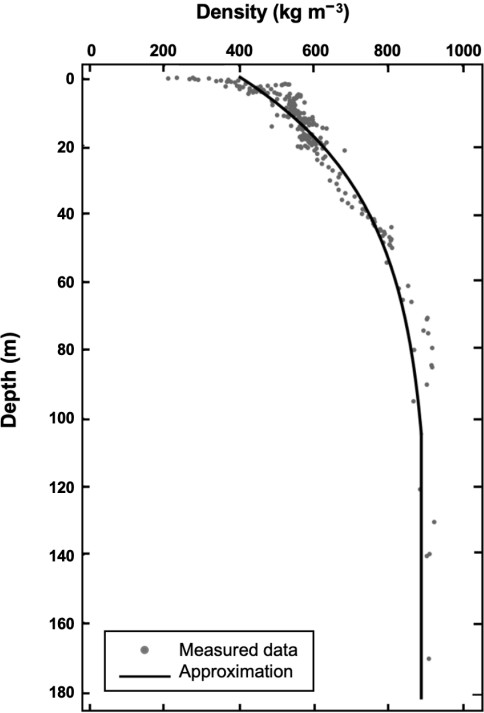

**Figure 3.** Vertical distribution of ice/firn density. The dots mark the density values determined from the 2009 ice core; the line shows the graph of the function (Eq. (4)) approximating these values.

The functions of the relative density $\varphi$ ($\varphi = \rho/\rho_i$)

$$a(\varphi) = \begin{cases} \exp\left(13.2224 - 15.78652\varphi\right), & \varphi \leq 0.81 \\ \frac{1}{3}\left(5 - 2\varphi\right)\varphi^{-\frac{2n}{n+1}}, & \varphi > 0.81 \end{cases} \tag{5}$$

and

$$b(\varphi) = \begin{cases} \exp\left(15.09371 - 20.46489\varphi\right), & \varphi \leq 0.81 \\ \frac{3}{4}\left(\dfrac{(1-\varphi)^{\frac{1}{n}}}{n\left(1-(1-\varphi)^{\frac{1}{n}}\right)}\right)^{\frac{2n}{n+1}}, & \varphi > 0.81 \end{cases} \tag{6}$$

taken from the work of Zwinger et al. (2007) are used to construct the flow law relations.

Following the work of Gagliardini and Meyssonnier (1997), we introduce the tensor invariant

$$\delta = \sqrt{\frac{2\operatorname{tr}\left(\mathbf{D}^{\mathrm{D}}\right)^2}{a\left(\varphi\right)} + \frac{(\operatorname{div}\boldsymbol{v})^2}{b\left(\varphi\right)}}.\tag{7}$$

Hereinafter $\operatorname{tr}$ denotes the trace of a tensor, the superscript D denotes the deviator of a tensor.

The melting temperature of ice is considered to be pressure-dependent: $T_{\mathrm{m}} = T_0 - \beta p$. The temperature relative to the pressure melting point is defined as follows: $T' = T - T_{\mathrm{m}} + T_0$ (Greve and Blatter, 2009).

In the case of thermomechanically coupled model, the rate factor $A$ is dependent on the temperature relative to the pressure melting point $T'$ and is determined by the Arrhenius law:

$$A\left(T'\right) = A_0 e^{-\frac{Q}{RT'}}.\tag{8}$$

Then the viscosity of ice/firn can be expressed as follows:

$$\mu\left(T', \delta, \varphi\right) = \frac{\left(2EA\left(T'\right)\right)^{-\frac{1}{n}} \delta^{\frac{1-n}{n}}}{a\left(\varphi\right)}.\tag{9}$$

The constant flow enhancement factor $E$ in Eq. (9) is used as a calibration parameter.

After decomposing the Cauchy stress tensor into an isotropic and a deviatoric parts

$$\boldsymbol{\sigma} = -p\mathbf{I} + \boldsymbol{\sigma}^{\mathrm{D}},\tag{10}$$

where $p = -\operatorname{tr}\boldsymbol{\sigma}/3$, we can write the constitutive relations for each component:

$$p = -\frac{a\left(\varphi\right)\mu}{b\left(\varphi\right)}\operatorname{div}\boldsymbol{v},\tag{11}$$

$$\boldsymbol{\sigma}^{\mathrm{D}} = 2\mu\mathbf{D}^{\mathrm{D}},\tag{12}$$

where $\mu$ is determined by Eq. (9).

The specific heat

$c\left(T\right) = 146.3 + 7.253T$                                                 (13)

and the heat conductivity

$$k\left(T\right) = 9.828 e^{-0.0057T}\tag{14}$$

of the ice/firn are temperature-dependent (Ritz, 1987).

The heat flux is determined by Fourier's law of heat conduction:

$\boldsymbol{q} = -k\operatorname{grad}T.$                                                            (15)

## 5.3 Field equations

The mechanical field equation of the model is the Stokes equation

$$\operatorname{div}\boldsymbol{\sigma} + \rho\boldsymbol{g} = \mathbf{0}. \tag{16}$$

In the case of thermomechanically coupled model the Stokes equation is supplemented with the heat transfer equation

$$\rho c\left(\frac{\partial T}{\partial t} + \boldsymbol{v}\cdot\operatorname{grad}T\right) = -\operatorname{div}\boldsymbol{q} + \operatorname{tr}\left(\boldsymbol{\sigma}\cdot\mathbf{D}\right), \quad T \leq T_{\mathrm{m}}. \tag{17}$$

Since the model is steady-state, $\partial T/\partial t \equiv 0$. The solution is limited by the pressure melting point $T_{\mathrm{m}}$.

## 5.4 Boundary conditions

Boundary conditions (BCs) are specified separately at three areas of the boundary of the computational domain: the surface, the base, and the lateral side.

### 5.4.1 Surface boundary conditions

Surface BCs apply to the part of the glacier surface under consideration (designated by the subscript s). They include the stress-free condition

$$\left.(\boldsymbol{\sigma}\cdot\boldsymbol{n})\right|_{\mathrm{s}} = \mathbf{0} \tag{18}$$

and the surface temperature condition

$$\left.T\right|_{\mathrm{s}} = T_{\mathrm{s}}. \tag{19}$$

The surface temperature value $T_{\mathrm{s}} = -18\,^{\circ}\mathrm{C}$ is selected on the basis of 2009 borehole temperature measurements ($-17.3\,^{\circ}\mathrm{C}$ at 10 m depth) and meteorologically based estimation of the annual mean air temperature at the drill site of $-19\,^{\circ}\mathrm{C}$ (Mikhalenko et al., 2015).

### 5.4.2 Basal boundary conditions

Basal BCs (denoted by b) apply to the bedrock of the glacier. The dynamical basal BC implies a cold base (zero tangential velocity) and a small normal outflow ice velocity at the bedrock:

$$\left.\boldsymbol{v}\right|_{\mathrm{b}} = v_{\mathrm{b}}\boldsymbol{n}|_{\mathrm{b}}. \tag{20}$$

Under the BC (20), ice/firn particles are modeled as moving from the glacier surface to the bedrock in finite time, which ensures convergence in the numerical solution of the dating problem (1)–(2). Also, such a small deviation of the basal velocity from zero apparently does not affect the dating results of the overlying ice/firn (except for a thin bottom layer), as indicated by the

coincidence of the age fields obtained with the $v_b$ increased and decreased by several orders of magnitude compared to the selected value.

The constant geothermal flux is specified as the thermodynamic bedrock BC:

$$(k \operatorname{grad} T \cdot \boldsymbol{n})|_b = q_{geo}. \tag{21}$$

The value of the basal heat flux $q_{geo} = 0.34 \text{ W m}^{-2}$ was calculated from the temperature measurements near the bottom of the 2009 borehole (Mikhalenko et al., 2015).

### 5.4.3 Lateral boundary conditions

Lateral BCs (denotation l) are specified at the vertical surface surrounding the modeled area and connecting the upper and lower parts of the computational domain boundary (glacier surface and bedrock). When modeling lateral outflow, the lateral BCs differ at the eastern (l, E) and western (l, W) sides of the domain, since the outflow is applied only at the western side (Fig. 1).

The WP glacier is bounded on the east by a steep wall of Elbrus summit, so we assume that there is no outflow from the domain through the eastern boundary:

$$(\boldsymbol{v} \cdot \boldsymbol{n})|_{l, E} = 0. \tag{22}$$

The western side of the plateau is characterized by westward slope, so outflow is expected at the western boundary of the domain. During field work, measurements of ice velocity on the glacier surface were not carried out. In this regard, we have performed all simulations for two extreme cases: 1) the outflow through the lateral side of the domain is completely absent; 2) the outflow is faster ($20 \text{ m a}^{-1}$ at the western rim of the surface) than would be expected in real conditions:

$$(\boldsymbol{v} \cdot \boldsymbol{n})|_{l, W} = 0 \quad \text{or} \quad (\boldsymbol{v} \cdot \boldsymbol{n})|_{l, W} = v_{out}, \tag{23}$$

where

$$v_{out}(d) = v_{max} \left( 1 - \left( \frac{d}{H} \right)^4 \right). \tag{24}$$

Here we assume that horizontal outflow decreases with depth in accordance with the forth power law. Similar relations are discussed in literature (e.g., Hooke, 2019; Greve and Blatter, 2009).

Zero normal heat flux is assumed at the entire lateral boundary:

$$(k \operatorname{grad} T \cdot \boldsymbol{n})|_l = 0. \tag{25}$$

The field equations of Sect. 5.3 together with the material equations of Sect. 5.2 and the BCs of Sect. 5.4 form a complete boundary value problem for determining the fields $\boldsymbol{v}$, $p$, and $T$.

## 6 Numerical methods

For finite element calculations, a flat computational grid (footprint) was created using the mesh generator Gmsh within the contour (shown in Fig. 1) of the area of the WP covered with DEMs. It contains 2508 nodes, 4734 linear triangular elements, and 280 linear one-dimensional elements on the boundary.

The volumetric computational grid was obtained by vertical extrusion of the flat grid. It is regular in the vertical direction (50 node levels) and characterized by a linear refinement of the step towards the glacier bed. The typical node spacing at the refined boundary is of order 0.1 m (less than 1 m), and at the coarser boundary it is of order 1 m (less than 10 m).

Differential field equations are solved numerically via their transformation to a discretized variational form (Gagliardini et al., 2013). In the numerical implementation, the constitutive relation (11) is interpreted as a field equation with unknowns $v$ and $p$.

For the thermomechanically coupled model the Stokes equation (16) (together with the Eq. (11)) and the heat transfer equation (17) are solved sequentially until convergence is achieved. On each step of this nonlinear iteration a system of linear algebraic equations arises and needs to be solved both for the Stokes equation and for the heat transfer equation. The linear systems for the Stokes equations are solved via an iterative method – the biconjugate gradient stabilized method (BiCGStab) whith zero-fill incomplete lower–upper preconditioner (ILU(0)). The linear systems for the heat transfer equation are solved via a direct method - unsymmetric-pattern multifrontal method, implemented in the solver UMFPACK.

In a purely mechanical case linear algebraic systems for the Stokes equation are solved via the same iterative method and preconditioner as in the coupled model.

The dating equation (1) is solved in a final step based on the velocity field calculated by the mechanical model. The equation was discretized via discontinuous Galerkin method and the resulting linear algebraic systems were solved by a direct method for the thermomechanically coupled model and by BiCGStab with ILU(1) preconditioner for the purely mechanical model.

## 7 Model calibration

All the simulation variants produced on the basis of the thermomechanically coupled model were also repeated by means of a purely mechanical model, i.e. a flow model with the heat transfer block switched off.

The simplification of the complete model is as follows. The ice/firn temperature is assumed to be constant ($-14$ °C) and close to the average value inferred from borehole measurements in 2009. In this case, the rate factor is constant ($2.291 \times 10^{-25}$ $\mathrm{Pa}^{-3}\,\mathrm{s}^{-1}$) and corresponds to this temperature (Cuffey and Paterson, 2010).

In order to provide the best fit between age–depth distributions simulated for the location $a$ and derived from empirical data for the location $b$ (Fig. 1), we varied three independent characteristics of the model. The first is the general type of the model (thermomechanically coupled or purely mechanical). The second is the presence or absence of lateral ice/firn outflow. The third characteristic is the viscosity value. Technically, the calibration of viscosity comes down to choice of the value of flow enhancement factor (see Eq. (9)).

Changing the enhancement factor results in a general increase/decrease in ice/firn velocities. The velocities, in turn, affect the age field and, in particular, the configuration of the age–depth curve at the studied location. For the purely mechanical model, we varied the enhancement factor in the range from 0.15 to 0.35; for the thermomechanically coupled model, in the range from 0.01 to 0.05. Choosing the value outside these ranges resulted in a significant discrepancy between the model and the empirical age–depth dependencies (for more details, see the Discussion section).

## 250   8  Results

The results presented below correspond to three selected simulation cases among numerous cases for each of which the ice-core-derived age–depth dependency is reproduced relatively well. These cases represent limiting scenarios, as the age–depth curves for the remaining simulations lie within the region bounded by the curves of these three cases. The combinations of model characteristics for these cases are presented in Table 2.

**Table 2.** Comparison of datings of reference horizons.

| $E$ | 0.016 | 0.2 | 0.25 | | |
|---|---|---|---|---|---|
| Lateral outflow | No | Yes | No | | |
| Model type | Thermomechanical | Mechanical | Mechanical | | |
| $d$ (m) | Model date (CE) | | | Proxy date (CE) | Proxy description |
| 75.6 | 1959 | 1966 | 1979 | 1963 | $^3$H |
| 116.7 | 1920 | 1913 | 1945 | 1912 | Katmai |
| 146.38 | 1866 | 1829 | 1886 | 1854 | Shiveluch |
| 153.7 | 1838 | 1785 | 1853 | 1815 | Tambora |
| 154.73 | 1831 | 1774 | 1845 | 1815 | Tambora |
| 160.4 | 1807 | 1734 | 1815 | 1783 | Laki |
| 177.11 | 1367 | 981 | 1288 | 670–1245 | $^{14}$C |
| 179.19 | 1021 | 386 | 877 | 130–770 | $^{14}$C |

Figure 4 illustrates the simulation results for a purely mechanical model without lateral outflow, showing the velocity magnitude and isochrones in the vertical section passing through the 2009 drilling site. The velocity of ice/firn in this section reaches its highest values (about $12 \, \mathrm{m \, a^{-1}}$) near the southern boundary of the computational domain, due to the tangential (to the vertical "wall") component. The lateral boundary condition in this simulated case sets the normal velocities to zero, while the tangential velocities remain unconstrained.

Figure 5 presents vertical profiles of modeled age and temperature of the ice/firn at a selected location shifted slightly southward from the drilling site. The modeling results are compared to the laboratory-derived 2009 ice core dating results (Fig. 5a) and the borehole temperature measurements from 2009 (Fig. 5b). The measured data are presented in interpolated form.

In Fig. 5a, the dating of the upper 168.6 m section of the ice core (Mikhalenko et al., 2024) is referred to as "Measured data". This is a revised version of the chronology by Preunkert et al. (2019), dating the deepest layers back to 1750. Additionally,

calibrated date ranges at a 68.2 % confidence level for radiocarbon dating (Preunkert et al., 2019) of four basal ice samples are shown. The depth ranges of the sampled ice are $177.11 \pm 0.22$ m, $179.19 \pm 0.14$ m, $181.50 \pm 0.13$ m, and $182.02 \pm 0.13$ m (Preunkert et al., 2019).

The ice in the intermediate part of the core (depth range 168.6–177.11 m) has not previously been dated by other methods. Using modeling, we obtained a probable dating range for each horizon in this part of the ice core. The lower bounds of these

date ranges correspond to a thermomechanically coupled model without lateral ice outflow and a flow enhancement factor of 0.016, while the upper bounds correspond to a purely mechanical model with lateral outflow and a flow enhancement factor of 0.2 (Fig. 5a). The lengths of these date ranges increase with depth from 130 years at 168.6 m to 386 years at 177.11 m. At a depth of 168.6 m, the modeled date range is 1591–1721 (the age is overestimated, as the corresponding date based on annual layer counting is approximately 1750). At 177.11 m, the modeled date range overlaps with the 68.2 % confidence interval of

the radiocarbon date (the overlapping range is 981–1245; see Table 2).

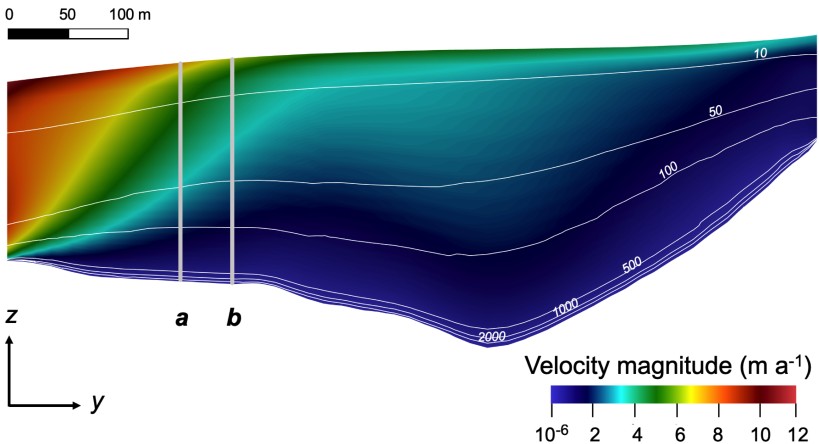

**Figure 4.** Cross-section of the computational domain by a vertical plane (case $E = 0.25$, $v_{\text{out}} = 0$). The cross-section is drawn in the north-south direction and passes through the drilling site of 2009 (vertical line $b$; the modeling results presented in Fig. 5 refer to vertical line $a$). The $y$ coordinate axis is directed north, $z$ – upwards. The colors of the rainbow show the velocity value (m/a). The white curves are isochrones corresponding to the model ice/firn ages of 10, 50, 100, 500, 1000, and 2000 years.

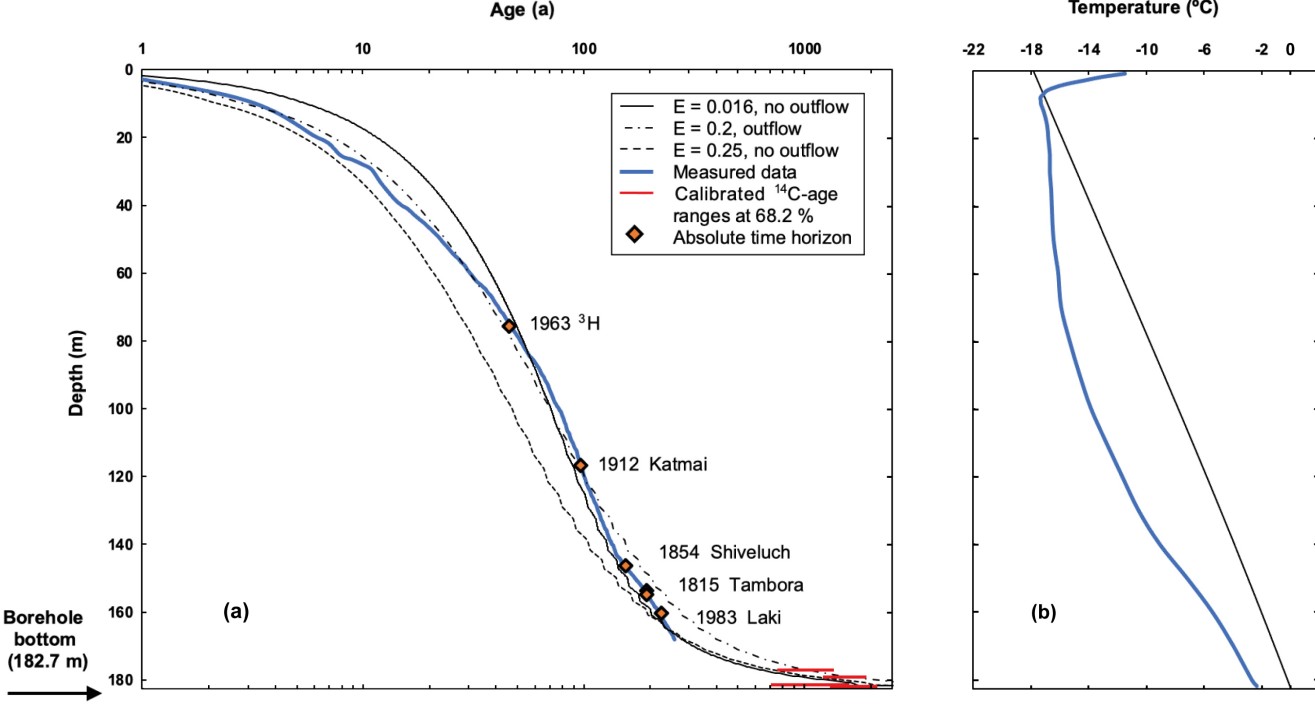

**Figure 5.** Modeled ice/firn age (a) and temperature (b) vertical profiles for the location *a* (see Fig. 4) and empirical data.

Table 2 compares the dates obtained from our simulations with the 2009 ice core dating for reference horizons and the two upper basal ice samples. The two lower ice samples are not included in Table 2 because the model significantly overestimates their ages. The age of the basal ice is formally equal to 100,000 years due to the upper limit specified in the numerical solution. Similarly to the vertical profiles in Fig. 5, the model results in the table correspond to location *a* rather than the drilling site *b*

(Fig. 1). The depth values for the tritium horizon ($^3$H), formed as a result of nuclear tests, and for volcanic horizons (Katmai, Shiveluch, Tambora, and Laki) provided based on the more reliable of the two main chronologies from the work of Mikhalenko et al. (2024). Note that Table 2 reflects the uncertainty in determining the depth of the Tambora volcanic layer. For the bottom ice samples, the table presents the median sample depths along with calibrated $^{14}$C date ranges at a 68.2 % confidence level, as reported by Preunkert et al. (2019).

The simulated temperature of the basal ice at the location *a* is equal to the pressure melting temperature for this point (−0.14 °C); the temperature measured at the bottom of the 2009 borehole is −2.4 °C (Mikhalenko et al., 2015).

## 9  Discussion

The modeling of the velocity field and the age of the ice in the WP glacier was conducted under several simplifying assumptions and limitations. The key assumptions are as follows:

1. *Stationarity:* This assumes a steady-state geometry of the glacier, along with unchanging density, velocity, pressure, and temperature fields.

2. *Lack of velocity data:* No direct measurements of the glacier velocity field are available.

3. *Surface temperature BC:* The temperature at the glacier surface reflects early 21st-century conditions and does not account for historical variations.

4. *Geothermal flux:* A locally determined geothermal flux value is extrapolated across the entire glacier bed.

5. *Density distribution:* The density distribution is not simulated and depends only on depth.

6. *Neglecting bubbles close-off:* The pressure in air bubbles after their close-off in the lower layers of firn is not taken into account when calculating the pressure field.

7. *Heat conductivity parameterization:* The dependence of thermal conductivity on density is not accounted for.

8. *No basal melting:* The model does not incorporate the potential effects of basal melting.

9. *Ice thickness uncertainty:* Ice thickness estimates have an uncertainty of several meters.

10. *Spatial mismatch:* The simulations performed for a site that does not match exactly the location of the 2009 drilling site.

Conditions 1–4 are determined by the lack of data on the modern and historical state of glaciation of the WP.

In the presence of millennial scale paleoclimatic data (air temperature and surface mass balance), it would be possible to
perform a transient simulation of glacier geometry and the long-term change of the flow regime, which could make the dating of basal ice formed under conditions different from the modern ones more accurate.

An alternative to the assumption of spatial homogeneity and stationarity of the geothermal flux (condition 4) could be some kind of its parameterization, for example, a power-law dependence on bedrock elevation, as in the work of Zwinger et al. (2007). However, the authors mentioned had several empirical values of the basal heat flux at the site of their study. We have a
single value obtained from field data, so any parameterization of the geothermal flow would remain speculative. The relatively high value of heat flux at the glacier bed on WP can be explained by the action of the magma chamber of the Elbrus volcano (Likhodeev and Mikhalenko, 2012; Mikhalenko et al., 2015). Also, in the study by Masurenkov (1971), a close value of heat flux ($0.347 \, \mathrm{W\,m^{-2}}$) was obtained for the Garabashi glacier (southern slope of Elbrus), and the maximum value for Elbrus is estimated at $2 \, \mathrm{W\,m^{-2}}$.

Condition 5 is a significant simplifying assumption. A more rigorous approach is to calculate the density field based on the solution of the continuity equation together with the Stokes equation. In 2018, two more ice cores were obtained on the WP

– 150.3 m and 119.8 m long. These cores are not dated, but the ice/firn density measurements showed close agreement with the density distribution in the 2009 ice core (Mikhalenko et al., 2020). This suggests that the actual deviations of the density values from the distribution used are moderate.

According to the known parameterization, the pressure in air bubbles in ice/firn depends on the atmospheric pressure, as well as the temperature and relative density both at the point under consideration and at close-off (Lüthi and Funk, 2000). Taking this effect into account in the model (condition 6) would lead to a change in the simulated pressure field, and consequently the velocity field and, ultimately, the age field. Considering bubbles close-off would provide an additional calibration parameter for more accurate reproduction of the empirical age–depth dependency, for example, by varying the close-off density, as

was done in the work of Lüthi and Funk (2000). According to the results of their study, when changing this parameter, the increase/decrease in the simulated age of the ice is more pronounced the greater the depth, which can be used to refine the age of near-bottom ice.

Including the dependence of heat conductivity on density in the model (condition 7) was tried but did not lead to adequate results. A more effective approach was found to be to compensate for this simplified parameterization by choosing appropriate

values of the flow enhancement factor.

The flow enhancement factor values selected as a result of model calibration turned out to be less than 1, which is not typical for glacier dynamics models (Greve and Blatter, 2009) and indicates that without appropriate correction the flow model overestimates the ice/firn fluidity. Similar results were reported previously for a crater glacier in Kamchatka where the value of the flow enhancement factor was also less than unity ($E = 1/3$) (Zwinger et al., 2007). Further analysis will be required to

identify the causes of the atypical shift in the value of this parameter.

Concerning condition 8, a study of the WP temperature regime (Mikhalenko et al., 2015), based on the method proposed by Salamatin et al. (2001), suggested basal melting occurs at depths exceeding 220 m, with rates not exceeding 10 mm water equivalent per year. Therefore, most of the glacier bed is probably cold, and we assume that the cold glacier model is generally adequate for the WP.

Condition 10 is a consequence of condition 9. The spatial arrangement of the isochrones (Fig. 4) indicates minimal differences in the ice/firn age distributions between verticals *a* and *b*. Thus, using the age distribution along vertical line *a* to calibrate the model provides a satisfactory representation of the broader area, including the 2009 core drilling site.

As highlighted, the region between the curves in Fig. 5a represents the range of age–depth relationships derived from the modeling, as these curves correspond to the limiting cases. From this perspective, the model reproduces the age of the ice for

the upper part of the core (down to a depth of about 165 m) quite accurately, overestimates the age in the depth range of 165–170 m, and aligns with the 68.2 % confidence intervals for the ages of the upper two of the four basal ice samples. However, steady-state models for cold glaciers are known to be unsuitable for dating the deepest ice layers near the bedrock (Zwinger et al., 2007). Therefore, our model could not capture the age confidence intervals for the two lowermost basal ice samples (Fig. 5a).

The vertical temperature distribution, simulated using the thermomechanically coupled model, is almost linear and overestimates temperatures (Fig. 5b). Thus, the mechanical coupling is weakly manifested.

## 10 Conclusions

A series of 3D ice/firn age distributions in the glacier on the Elbrus Western Plateau was obtained, using an ice flow model in thermomechanically coupled and purely mechanical implementations, varying boundary conditions and calibration parameter values. From all the simulation results, we determined the range of modeled age–depth curves corresponding to the 2009 ice core. The pattern of isochronous surfaces suggests that the obtained ages of the ice are generally representative of the area near the 2009 drilling site. The dating results are robust to small variations in the calibration parameters and align closely with the dating of the upper part of the 2009 ice core. The correspondence extends to most of the core (1766 CE and later). In the depth range about 3.5–5.6 m above the bedrock, all model dates (386–1367 CE) fall within or near the 68.2 % confidence intervals of radiocarbon-dated ice core samples. Comparison with ice core data supports the validity of the model dating except for the 3–4 m bottom layer. Additionally, the model provided an approximate reconstruction of the previously undated middle part of the 2009 ice core. To our knowledge, we have constructed the first 3D age model for the glacier on the Elbrus Western Plateau. The model may become the basis for identifying the most promising sites for ice core drilling on the plateau in the future.

. *Data availability.* Input data for Elbrus Western Plateau (Caucasus) ice flow model are available at https://doi.org/10.5281/zenodo.14795959 (Chernyakov et al., 2025).

. *Author contributions.* GC performed the simulations and prepared the manuscript with contributions from all co-authors, NE performed visualization of the data and results, TK provided the study area description and mapping, SK contributed to conceptualization and writing.

. *Competing interests.* The authors declare that they have no conflict of interest.

. *Acknowledgements.* The work of Gleb Chernyakov, Nelly Elagina, and Taisiia Kiseleva was carried out within the framework of the Russian Science Foundation project no. 24-27-00262. The Pléiades DEM used in this study was provided by the Pléiades Glacier Observatory initiative of the French Space Agency (CNES). The authors thank Olivier Gagliardini and Vladimir Mikhalenko for fruitful discussions that significantly contributed to the work. We are also grateful to Martin Lüthi and the anonymous reviewer for constructive comments that helped to considerably improve the manuscript.

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
