# Peer review of "Ice/firn age distribution on the Elbrus Western Plateau (Caucasus) inferred from ice flow model"

_EGUsphere, 2024_

## Referee Comment (RC1)

**General comments**

The paper "Ice/firn age distribution on the Elbrus Western Plateau (Caucasus) inferred from ice flow model" by Gleb Chernyakov, Nelly Elagina, Taisiia Kiseleva, and Stanislav Kutuzov presents a modeling study that employs a thermo-mechanical ice/firn flow model to analyze the age-depth relationship of an ice core drilled in 2009 on the Western Plateau of Mount Elbrus (Caucasus). The study explores multiple model runs, varying the outflow boundary conditions (BC), ice viscosity, and the inclusion of thermodynamical coupling. Among these runs (which are not all presented), the authors identify three configurations that are said to define an envelope within which all other simulated age-depth curves lie. A figure seems to show that the three selected cases yield age-depth relationships consistent with available measurements. However, these measurements are represented as a continuous line, whereas discrete data points would be expected (see below).

Regarding novelty, this study is not the first of its kind and appears rather minimalistic compared to previous works (e.g., Zwinger et al., 2007; Gilbert et al., 2014; Licciulli et al., 2019). The only aspect that distinguishes it from existing studies is the choice of the study site.

Regarding the methodology, the thermodynamical model appears to be quite simplistic. It is formulated in terms of temperature rather than enthalpy, effectively excluding the possibility of melting/refreezing. This assumption may be reasonable given the stated "negligible seasonal melting," but it should at least be explicitly mentioned and justified. In addition, the model neglects the dependence of firn thermal conductivity on density, as evident from Eq. (15) and the linear temperature profile shown in Fig. 4c. This is a questionable assumption, and studies such as Zwinger et al. (2007), Gilbert et al. (2014), and Licciulli et al. (2019) have all accounted for this dependency. The surface Dirichlet boundary condition (BC) is set at a fixed temperature of $-18$ C, which is acceptable given that the simulations are steady-state. Similarly, the bottom Neumann BC assumes a uniform geothermal heat flux over the entire modeled domain, which is reasonable given the lack of data. At least, these two last assumptions are (very briefly) discussed in the last Section of the paper.

Overall, the relevance of this thermodynamical model is unclear. The only two-way coupling with the mechanical model would be through the advection term, yet the simulated temperature profile remains linear, suggesting that this coupling has little to no effect. By the way, I am surprised by this linear profile. Given the temperature-dependent thermal conductivity specified in Eq. (15), I would expect at least some degree of non-linearity in the temperature profile. Am I wrong ?

Apart from the weak constraints on outflow velocities, the mechanical model appears reasonable, with one major exception: the mass conservation equation is never solved. This implicitly assumes that the 2009 density distribution applies uniformly across the entire modeled domain. Again, this is a strong assumption that should be explicitly discussed. A more rigorous approach would be to solve the Stokes equation and the mass conservation equation sequentially until a steady density profile is obtained, as done by, e.g., Gilbert et al. (2014) or Brondex et al. (2020). Naturally, given the uncertainties in the parameterization of functions $a(\phi)$ and $b(\phi)$, the resulting density profile would likely deviate somewhat from the measured 2009 distribution (Brondex et al., 2020). Again, I am surprised by the velocity field magnitude shown in Fig. 3. If I understand correctly, this result corresponds to the simulation where the outflow lateral boundary condition imposes zero normal velocity. If that is the case, how can such high (relatively) velocities be observed at the southern boundary ?

Another point regarding the methodology is the choice to evaluate the simulated profile at a location 50 m away from the actual drilling site. While I am fairly certain this does not significantly affect the results, I find the justification somewhat unclear. First, the fact that there is only a $\sim 6$ m discrepancy between the length of the core and the ice thickness evaluated by differencing the bottom and surface DEMs is relatively good, especially considering the precision typically achievable with GPR measurements for an ice thickness of around $\sim 180$ m. Second, while I am not an expert in remote sensing, I would expect that, although the mean annual surface elevation has likely been stable in recent years, there could be significant sub-annual variability in surface elevation, which could also explained part of the discrepancy. The surface DEM used to construct the computational domain seems to be from 2017, but was it retrieved at the end of summer to match the period when the drilling was done? In my opinion, instead of shifting the point of interest by 50 m, it would have been more appropriate to verify that the altitude provided by the DEM at the drilling site aligns closely with the field measurements taken in 2009. My general point is that, ideally, the depth 0 of the core should correspond as closely as possible to the surface altitude of the computational domain.

Regarding the results, I have the impression that the figures are presented to the reader without being adequately described in the text. For instance, the purpose of showing velocity magnitudes in Fig. 4a is unclear, except perhaps to illustrate that velocities are essentially vertical, as suggested by the quasi-symmetry of the figure. However, this is not discussed anywhere in the text. Additionally, I don't understand why the measured ages are plotted as a continuous line in Fig. 4b, while I would expect discrete points corresponding to the ages listed in the "Proxy date"

column of Table 2. One of the motivations of the study is stated as being to "reconstruct the age of the intermediate section of the 2009 core, which had not previously been dated by other methods." However, this outcome is not at all evident from this figure.

The discussion is also quite limited. The main assumptions of the model are only briefly addressed, with minimal justification provided (and some assumptions are even omitted, as noted earlier). There is a short mention of basal ice ages that "fall within or near the 68.2% confidence intervals of radiocarbon-dated ice core samples." However, given the large uncertainties associated with both the radiocarbon dating and the model at these depths, it is difficult to draw any meaningful conclusions. Additionally, a discussion on the "calibration procedure" would have been appreciated. As it stands, one might get the impression that the authors ran thousands of simulations, discarded 997 of them, and kept only the three that gave the best fit to the available measurements. While I am sure this is not the case (since the authors mention that the three selected simulations bound the ensemble of depth-age curves) more information on this process would have been useful.

Otherwise, the English is good, and the quality of the figures is relatively good. However, many citations are presented in parentheses when they should not be, especially towards the end of the introduction.

In my opinion, the paper cannot be published in its current form, and major revisions are required. Below, I list my specific comments.

**Specific comments**

p2 L26 It would be helpful to mention whether the 2009 drilling reached the bed. We understand later on that it did but mention it from the introduction would be preferable.

p2 L34-35 It is somewhat unclear what is meant by 'basal ice' and 'deepest ice'. I understood 'basal ice' to refer to the ice located at the bottom of the drill site, whereas 'deepest ice' seems to refer to thicker ice found elsewhere on the plateau, where the glacier is deeper. Clarifying this distinction would help improve readability.

p2 L47-59 Many citations are given in parentheses when they should not. Please check and correct here and elsewhere in the manuscript.

p2 L49-50 Remove "Mt" and replace "Mont Blank" by "Mont Blanc".

p2 L50-51 Gagliardini and Meyssonnier (1997) are actually dealing with the same study site as Vincent et al. (1997).

p2 L50-51 Gagliardini and Meyssonnier (1997) are actually dealing with the same study site as Vincent et al. (1997).

p2 L50-51 Similarly, Licciulli et al. (2019) actually investigate the same study site as Lüthi and Funk (2000). I would suggest restructuring the paragraph to create a clearer logical flow. Currently, it shifts from one study site to another, then circles back to the first site. An alternative approach could be to structure the paragraph chronologically, highlighting the progress made over the years from basic models to increasingly complex ones. I believe this might be your intention, but at present, it does not come across clearly.

p2 L62 I guess it is the yearly averaged isotherm (averaged over which period ?).

p2 L73 The year and the time of year when the DEM was produced should be specified here, as this information is important for context and interpretation.

p3 L78-79 It is not entirely clear whether both the surface and bedrock DEMs are from 2017. As currently written, it seems that only the radar surveys, and therefore the bedrock DEM, were conducted in 2017.

p4 Fig. 1 Even if explained later, I believe the definitions of 'a' and 'b' should be specified directly in the caption. I would also recommend explicitly mentioning that the top-right figure is a zoom of the black dashed square.

p4 L92 I think it is always helpful to specify the typical node spacing at both the refined boundary and the coarser boundary when working with refined meshes.

p5 L93-98 I find this argumentation confusing. See my general comments.

p5 L100-101 I would recall that there is an assumption of steady state.

p7 L116 "Next, the constitutive equations of the model are subsequently introduced" $\rightarrow$ I would remove this sentence.

p8 L126 For $a$ and $b$, you are not using the original parameterization proposed by Gagliardini and Meyssonnier (1997) but the corrected ones proposed by Zwinger et al. (2007) and then re-used succesfully by, e.g., Gilbert et al. (2014), Licciulli et al. (2019) or Brondex et al. (2020).

p8 L136 You forgot to mention what is $T_0$.

p9 L13 and below The way you are presenting the constitutive laws/field equations is kind of strange to me. Equation (13) alone is sufficient for pure ice, as it reduces to the usual Glen's law when $\rho = \rho_i$. For pure ice, the incompressibility assumption leads to div $\mathbf{v} = 0$, which implies that the strain rate tensor is purely deviatoric. However, when dealing with compressible firn, an additional equation for the spheric parts of the stress (i.e., $p$) and strain rate (i.e., div $\mathbf{v}$) tensors is required to close the constitutive relationship. This equation is:

$$p = -\frac{1}{b}(2A)^{-\frac{1}{n}}\gamma^{\frac{1-n}{n}} \operatorname{div} \mathbf{v}. \tag{1}$$

I agree that combining this equaiton to your Eq. (11) results in your Eq. (17), but in my view, this is a constitutive law (a relationship between stresses and strain rates) rather than a field equation. I am aware that Zwinger et al. (2007) also presents your Eq. (17) as a field equation. Conversely, the missing field equation in your formulation, in my opinion, is the mass conservation equation:

$$\frac{\partial \rho}{\partial t} + \operatorname{div} \rho \mathbf{v} = 0. \tag{2}$$

The way the problem is presented implies that the unknowns are $(u, v, w, p)$ and the corresponding system of four scalar equations is given by the three scalar equations in your Eq. (18) together with the scalar Eq. (17). However, this formulation neglects the fact that $\rho$ is another unknown, and thus Eq. (2) is necessary to close the system. Assuming that $\rho$ is known from core measurements and can be applied uniformly across the domain is a strong assumption that should at least be mentioned and discussed (see my general comments).

p9 L149 A reference to justify this parameterizaiton would be welcome.

p9 L151 Same as above. I also find it strange that the dependence of the thermal conductivity of firn on its density is not accounted for (see my general comments).

p9 L161 A thermodynamic model that operates in steady state, does not account for variations in thermal conductivity with density, and is unable to handle melting and refreezing (as shown from the fact that it is expressed in terms of temperature instead of enthalpy and lacks a latent heat source/sink term) calls into question its relevance for the present study. Please refer to my general comments.

p10 L179 I don't get this boundary condition. Normally at the bed, a non-penetration condition applies and the normal velocity is forced to zero.

p11 L205-211 I am not sure this explanation desserves a full paragraph and I think that the last sentence could be removed.

p11 L218-220 The description of the numerical implementation is a bit unclear. From my understanding, the mechanical and thermodynamic problems are solved sequentially using a first-operator splitting approach until convergence is achieved (referred to as the 'steady state iterations' in Elmer). Independently of this coupling, the Stokes equation (Porous Solver in Elmer) is non-linear due to $n = 3$ in the constitutive law and needs to be linearized. The same approach applies to the heat equation, which is non-linear due to the $T$-dependence of heat capacity and thermal conductivity. In each non-linear iteration, a linear system is obtained, which can be solved using either direct or iterative methods.

p12 L226 "a precalculated velocity field" $\rightarrow$ does that refer to the velocity field calculated by the mechanical model (I guess it does) ? Please, clarify.

p13 L248 The last two samples do not appear in Table 2. Why ?

p16 L282-283 This is a bit of an overstatement. Your modelled temperature profile is actually quite far from the measured one.

**References**

Brondex, J., Gagliardini, O., Gillet-Chaulet, F., and Chekki, M.: Comparing the long-term fate of a snow cave and a rigid container buried at Dome C, Antarctica, Cold Regions Science and Technology, 180, 103164, https://doi.org/10.1016/j.coldregions.2020.103164, 2020.

Gagliardini, O. and Meyssonnier, J.: Flow simulation of a firn-covered cold glacier, Annals of Glaciology, 24, 242–248, https://doi.org/10.1017/S0260305500012246, 1997.

Gilbert, A., Gagliardini, O., Vincent, C., and Wagnon, P.: A 3-D thermal regime model suitable for cold accumulation zones of polythermal mountain glaciers, Journal of Geophysical Research (Earth Surface), 119, 1876–1893, https://doi.org/10.1002/2014JF003199, 2014.

Licciulli, C., Bohleber, P., Lier, J., Gagliardini, O., Hoelzle, M., and Eisen, O.: A full Stokes ice-flow model to assist the interpretation of millennial-scale ice cores at the high-Alpine drilling site Colle Gnifetti, Swiss/Italian Alps, Journal of Glaciology, pp. 1–14, https://doi.org/https://doi.org/10.1017/jog.2019.82, 2019.

Lüthi, M. and Funk, M.: Dating of ice cores from a high Alpine glacier with a flow model for cold firn, Annals of Glaciology, 31, 69–79, https://doi.org/10.3189/172756400781820381, 2000.

Vincent, C., Vallon, M., Pinglot, J. F., Funk, M., and Reynaud, L.: Snow accumulation and ice flow at Dôme du Goûter (4300 m), Mont Blanc, French Alps, Journal of Glaciology, 43, 513–521, https://doi.org/10.3189/S0022143000035127, 1997.

Zwinger, T., Greve, R., Gagliardini, O., Shiraiwa, T., and Lyly, M.: A full Stokes-flow thermo-mechanical model for firn and ice applied to the Gorshkov crater glacier, Kamchatka, Annals of Glaciology, 45, 29–37, https://doi.org/10.3189/172756407782282543, 2007.

---

## Author Comment (AC1)

**Reply on Referee Comment 1**

Dear Reviewer,

We are very grateful for the work you have done, and we sincerely appreciate your thorough reading of our manuscript and provided fair and constructive comments. In the response below, your review comments are shown in blue, our responses are in black, and the corrected or added parts of the manuscript are highlighted in purple.

**General comments**

The paper "Ice/firn age distribution on the Elbrus Western Plateau (Caucasus) inferred from ice flow model" by Gleb Chernyakov, Nelly Elagina, Taisiia Kiseleva, and Stanislav Kutuzov presents a modeling study that employs a thermo-mechanical ice/firn glow model to analyze the age-depth relationship of an ice core drilled in 2009 on the Western Plateau of Mount Elbrus (Caucasus). The study explores multiple model runs, varying the outflow boundary conditions (BC), ice viscosity, and the inclusion of thermodynamical coupling. Among these runs (which are not all presented), the authors identify three configurations that are said to define an envelope within which all other simulated age-depth curves lie. A figure seems to show that the three selected cases yield age-depth relationships consistent with available measurements.

However, these measurements are represented as a continuous line, whereas discrete data points would be expected (see below).

Regarding novelty, this study is not the first of its kind and appears rather minimalistic compared to previous works (e.g., Zwinger et al., 2007; Gilbert et al., 2014; Licciulli et al., 2019). The only aspect that distinguishes it from existing studies is the choice of the study site.

> Indeed, we did not set out to develop new methods. We were mainly interested in the object. This work is the next step in a long-term series of studies of the glaciation of Elbrus and, in particular, the Western Plateau.

Regarding the methodology, the thermodynamical model appears to be quite simplistic. It is formulated in terms of temperature rather than enthalpy, effectively excluding the possibility of melting/refreezing. This assumption may be reasonable given the stated "negligible seasonal melting," but it should at least be explicitly mentioned and justified.

> The text has been revised accordingly (see our response to specific comment p2 L62). Also, in the "Discussion" section we have already noted the following:
>
> [...] a study of the WP temperature regime (Mikhalenko et al., 2015), based on the method proposed by Salamatin et al. (2001), suggested basal melting occurs at depths exceeding 220 m, with rates not exceeding 10 mm water equivalent per year.

In addition, the model neglects the dependence of firn thermal conductivity on density, as evident from Eq. (15) and the linear temperature profile shown in Fig. 4c. This is a questionable assumption, and studies such as Zwinger et al. (2007), Gilbert et al. (2014), and Licciulli et al. (2019) have all accounted for this dependency.

> We discuss this issue in response to specific comment p9 L161.

The surface Dirichlet boundary condition (BC) is set at a fixed temperature of −18 C, which is acceptable given that the simulations are steady-state. Similarly, the bottom Neumann BC assumes a uniform

geothermal heat flux over the entire modeled domain, which is reasonable given the lack of data. At least, these two last assumptions are (very briefly) discussed in the last Section of the paper.

Overall, the relevance of this thermodynamical model is unclear. The only two-way coupling with the mechanical model would be through the advection term, yet the simulated temperature prole remains linear, suggesting that this coupling has little to no effect. By the way, I am surprised by this linear prole. Given the temperature-dependent thermal conductivity specified in Eq. (15), I would expect at least some degree of non-linearity in the temperature prole. Am I wrong ?

In fact, this profile is slightly nonlinear, it's just that its nonlinearity is very small. We agree that thermomechanical coupling has little effect in our simulation. In general, it was just one of a series of numerical experiments. In our experience, in some cases, using a purely mechanical model to estimate the age of ice yielded better results in terms of matching ice core chronology, compared with the case of thermomechanical coupling. E.g., when scaling the rheological function in a coupled model, large corrections were required.

Apart from the weak constraints on outflow velocities, the mechanical model appears reasonable, with one major exception: the mass conservation equation is never solved. This implicitly assumes that the 2009 density distribution applies uniformly across the entire modeled domain. Again, this is a strong assumption that should be explicitly discussed. A more rigorous approach would be to solve the Stokes equation and the mass conservation equation sequentially until a steady density profile is obtained, as done by, e.g., Gilbert et al. (2014) or Brondex et al. (2020). Naturally, given the uncertainties in the parameterization of functions a(φ) and b(φ), the resulting density profile would likely deviate somewhat from the measured 2009 distribution (Brondex et al., 2020).

We agree that this issue deserves special attention. We have added the following to the "Discussion" section:

Condition 5 is a significant simplifying assumption. A more rigorous approach is to calculate the density field based on the solution of the continuity equation together with the Stokes equation. In 2018, two more ice cores were obtained on the WP – 150.3 m and 119.8 m long. These cores are not dated, but the ice/firn density measurements showed close agreement with the density distribution in the 2009 ice core (Mikhalenko et al., 2020). This suggests that the actual deviations of the density values from the distribution used are moderate.

Again, I am surprised by the velocity field magnitude shown in Fig. 3. If I understand correctly, this result corresponds to the simulation where the outflow lateral boundary condition imposes zero normal velocity. If that is the case, how can such high (relatively) velocities be observed at the southern boundary ?

This is due to the relatively high tangential (to the vertical 'wall') velocities at the southern boundary. The lateral boundary condition in this simulated case sets the normal velocities to zero, while the tangential velocities remain unconstrained.

Another point regarding the methodology is the choice to evaluate the simulated profile at a location 50 m away from the actual drilling site. While I am fairly certain this does not significantly affect the results, I find the justification somewhat unclear.

We discuss this issue in response to specific comment p5 L93-98.

First, the fact that there is only a ∼ 6 m discrepancy between the length of the core and the ice thickness evaluated by differencing the bottom and surface DEMs is relatively good, especially considering the precision typically achievable with GPR measurements for an ice thickness of around ∼ 180 m. Second, while I am not an expert in remote sensing, I would expect that, although the mean annual surface elevation

has likely been stable in recent years, there could be significant sub-annual variability in surface elevation, which could also explained part of the discrepancy.

Thank you for these ideas. In general, as noted below, the data obtained in different years relate to approximately the same season (July–September).

The surface DEM used to construct the computational domain seems to be from 2017, but was it retrieved at the end of summer to match the period when the drilling was done?

Yes, it matches quite well. The drilling was carried out in in August–September 2009, and the surface DEM is from 8 September 2017. See also our response to p2 L73.

In my opinion, instead of shifting the point of interest by 50 m, it would have been more appropriate to verify that the altitude provided by the DEM at the drilling site aligns closely with the field measurements taken in 2009. My general point is that, ideally, the depth 0 of the core should correspond as closely as possible to the surface altitude of the computational domain.

The difference between the elevation obtained using the 2017 DEM at the drilling site (5135.42 m above sea level) and the elevation determined during deep drilling in 2009 (5115 m above sea level; Mikhalenko et al., 2015) appears to be large, probably due to an assumed error in determining the latter. This makes comparison difficult.

Regarding the results, I have the impression that the figures are presented to the reader without being adequately described in the text. For instance, the purpose of showing velocity magnitudes in Fig. 4a is unclear, except perhaps to illustrate that velocities are essentially vertical, as suggested by the quasi-symmetry of the figure. However, this is not discussed anywhere in the text.

Indeed, the purpose of presenting this plot was to show how the vertical and the horizontal components of velocity are related, as well as typical magnitudes of velocity. We agree that this plot is not very informative in the context of our work and have decided to remove it. Also, in accordance with the recommendation of Reviewer 2, we presented the age dependence plot on a logarithmic scale and marked the reference horizons on it:

[Figure]

Additionally, I don't understand why the measured ages are plotted as a continuous line in Fig. 4b, while I would expect discrete points corresponding to the ages listed in the "Proxy date" column of Table 2.

This is mainly due to interpolation used in plotting the measured age (as well as the measured temperature). We have added a mention of interpolation to the text. In addition, the chemical dating of the upper part of the core has a resolution of about a year, and due to densification of firn at relatively large depths, neighboring dated points are close to each other, which even without interpolation gives the visual effect of a solid line in the age–depth plot.

One of the motivations of the study is stated as being to "reconstruct the age of the intermediate section of the 2009 core, which had not previously been dated by other methods." However, this outcome is not at all evident from this figure.

Agree. We have tried to fill this gap by adding the following description to the text:

The age of the ice in the intermediate part of the core (depth range 168.6–177.11 m), which has not been previously dated by other methods, can be estimated using modeling. For each horizon in this section, the model yields an age range depending on the selected parameters. The lower bound corresponds to a thermomechanically coupled model without lateral ice outflow and a flow enhancement factor of 0.016, while the upper bound corresponds to a purely mechanical model with lateral outflow and a flow enhancement factor of 0.2 (Fig. 5a). The difference between these estimates increases with depth, from 130 years at 168.6 m to 386 years at 177.11 m. At a depth of 168.6 m, the modeled age range is 1591–1721 (likely underestimated, as the corresponding age based on annual layer counting is approximately 1750). At 177.11 m, the modeled age range overlaps with the 68.2% confidence interval of the radiocarbon date (the overlapping range is 981–1245; see Table 2). It should also be noted that these age estimates refer to a location offset from the actual drilling site.

The discussion is also quite limited. The main assumptions of the model are only briefly addressed, with minimal justification provided (and some assumptions are even omitted, as noted earlier).

We have expanded the "Discussion" section and added the following points to the list of assumptions:

- *Density distribution:* The density distribution is not simulated and depends only on depth.
- *Heat conductivity parameterization:* The dependence of thermal conductivity on density is not accounted for.

There is a short mention of basal ice ages that "fall within or near the 68.2% confidence intervals of radiocarbon-dated ice core samples." However, given the large uncertainties associated with both the radiocarbon dating and the model at these depths, it is difficult to draw any meaningful conclusions.

Additionally, a discussion on the "calibration procedure" would have been appreciated. As it stands, one might get the impression that the authors ran thousands of simulations, discarded 997 of them, and kept only the three that gave the best fit to the available measurements. While I am sure this is not the case (since the authors mention that the three selected simulations bound the ensemble of depth-age curves) more information on this process would have been useful.

To describe the calibration procedure in more detail, we have added the following to the text:

Changing the enhancement factor results in a general increase/decrease in ice/firn velocities. The velocities, in turn, affect the age field and, in particular, the configuration of the age–depth curve at the studied location. For the purely mechanical model, we varied the enhancement factor in the range from 0.15 to 0.35; for the thermomechanically coupled model, in the range from 0.01 to 0.05. Choosing the value outside these ranges resulted in a significant discrepancy between the model and the empirical age–depth dependencies.

Otherwise, the English is good, and the quality of the figures is relatively good.

However, many citations are presented in parentheses when they should not be, especially towards the end of the introduction.

Corrected.

In my opinion, the paper cannot be published in its current form, and major revisions are required. Below, I list my specific comments.

**Specific comments**

p2 L26 It would be helpful to mention whether the 2009 drilling reached the bed. We understand later on that it did but mention it from the introduction would be preferable.

We agree with your comment and have added the following clarifications to the text:

Glaciochemical investigations of a deep ice core **(182.65 m long – from the glacier surface to solid rock)** drilled in 2009 include […].

[…]

According to DEMs ice thickness at the 2009 drilling site location is 6.1 m greater than ice core length. **Possible reasons for this discrepancy include: time difference in obtaining the DEMs of the surface and the base (8 years); errors in constructing the DEMs; inaccuracy in determining the coordinates of the drilling site; probable presence of a solid inclusion in the ice above the glacier bed, which could become a mechanical obstacle to drilling to the bed (Stiévenard et al., 1996).**

p2 L34-35 It is somewhat unclear what is meant by 'basal ice' and 'deepest ice'. I understood 'basal ice' to refer to the ice located at the bottom of the drill site, whereas 'deepest ice' seems to refer to thicker ice found elsewhere on the plateau, where the glacier is deeper. Clarifying this distinction would help improve readability.

We call the ice adjacent to the glacier bed 'basal' (regardless of its depth). By 'deepest ice' we mean the basal ice in the area of the glacier with the greatest ice thickness. To improve readability, we have changed the text as follows:

According to an estimate based on a two-dimensional (2D) analytical model of Salamatin et al. (2000), the age of **the ice at the bottom of the drilling site** does not exceed 350–400 years and the age of **the basal ice in the deepest part of the glacier** (more than 250 m deep) is about 660 years (Mikhalenko et al., 2015).

p2 L47-59 Many citations are given in parentheses when they should not. Please check and correct here and elsewhere in the manuscript.

Done.

p2 L49-50 Remove "Mt" and replace "Mont Blank" by "Mont Blanc".

Corrected.

p2 L50-51 Gagliardini and Meyssonnier (1997) are actually dealing with the same study site as Vincent et al. (1997).

Agree.

p2 L50-51 Similarly, Licciulli et al. (2019) actually investigate the same study site as Lüthi and Funk (2000). I would suggest restructuring the paragraph to create a clearer logical flow. Currently, it shifts from one study site to another, then circles back to the first site. An alternative approach could be to structure the

paragraph chronologically, highlighting the progress made over the years from basic models to increasingly complex ones. I believe this might be your intention, but at present, it does not come across clearly.

Indeed, we intended to describe the progress in model development. Text revised:

Ice flow models have been repeatedly used to study the age distribution of ice in mountain glaciers and, in particular, ice cores. **Dating models have undergone a natural evolution – from 2D analytical solutions to 3D numerical approaches that take into account the mutual influence of mechanical and thermal processes. The main stages of this development appear to us as follows.** 2D purely mechanical (Vincent et al., 1997) and thermomechanically coupled (Salamatin et al., 2000; Shiraiwa et al., 2001) analytical models were developed […].

p2 L62 I guess it is the yearly averaged isotherm (averaged over which period ?).

Text revised:

Glaciers cover an altitudinal range from 2680 to 5642 m a.s.l. **Above 5200 m a.s.l. temperature stays negative throughout a year and no melting occurs (Mikhalenko et al., 2020).**

p2 L73 The year and the time of year when the DEM was produced should be specified here, as this information is important for context and interpretation.

We have added the date of the Pléiades image to the text:

For the information about topography, we used the Pléiades digital elevation model (DEM) from **8 September 2017** with the vertical uncertainty between ±0.5 m and ±1 m (Kutuzov et al., 2019b).

p3 L78-79 It is not entirely clear whether both the surface and bedrock DEMs are from 2017. As currently written, it seems that only the radar surveys, and therefore the bedrock DEM, were conducted in 2017.

Text revised:

1. DEMs of the glacier surface and bedrock with a cell size of 10 × 10 m, both obtained in 2017.

p4 Fig. 1 Even if explained later, I believe the definitions of 'a' and 'b' should be specified directly in the caption. I would also recommend explicitly mentioning that the top-right figure is a zoom of the black dashed square.

Agree. The caption is corrected as follows:

**Figure 1.** Study area on Mt. Elbrus. **The vertical distributions obtained in our study (Fig. 5) refer to location *a*; the 2009 drilling site is marked with *b*. The top-right figure is a zoom of the black dashed square.** The elevation bands are based on the Pléiades DEM of 2017. The coordinates are presented in the WGS 84 (UTM 38 N). The SPOT 7 image obtained on 20 August 2016 is shown as a background.

p4 L92 I think it is always helpful to specify the typical node spacing at both the refined boundary and the coarser boundary when working with refined meshes.

Agree. The following addition has been made to the text:

The typical node spacing at the refined boundary is of order 0.1 m (less than 1 m), and at the coarser boundary it is of order 1 m (less than 10 m).

p5 L93-98 I find this argumentation confusing. See my general comments.

We have expanded the argumentation:

In our opinion, comparison of age–depth distributions (one modeled and one ice core based) for locations at a glacier with different ice thicknesses makes little sense. **Also, the glacier surface elevation at the drilling site (5135.42 m a.s.l.) obtained using the 2017 DEM does not align closely with the glacier surface elevation at the drilling site determined during deep drilling in 2009 (5115 m a.s.l.; Mikhalenko et al., 2015), probably due to an assumed error in**

**determining the latter. This makes problematic another possible interpretation of the original data, namely, matching the zero depth of the ice core with the surface altitude of the computational domain.**

To represent the vertical profiles of ice/firn age and temperature and compare them with ice core data, we chose one of the points closest to the drilling site with similar ice thickness (182.67 m according to DEMs) and topography of surface and bedrock.

p5 L100-101 I would recall that there is an assumption of steady state.

Agree. We have added this to the phrase:

**Under the assumption of steady state,** the velocity distribution in the glacier allows one to calculate the time required for each ice/firn particle to move from the glacier surface to its current position.

p7 L116 "Next, the constitutive equations of the model are subsequently introduced" → I would remove this sentence.

Agree. The sentence has been removed.

p8 L126 For *a* and *b,* you are not using the original parameterization proposed by Gagliardini and Meyssonnier (1997) but the corrected ones proposed by Zwinger et al. (2007) and then re-used succesfully by, e.g., Gilbert et al. (2014), Licciulli et al. (2019) or Brondex et al. (2020).

We are very grateful to the referee for pointing out the inaccuracy we made here. Now, when introducing the functions *a* and *b,* a more appropriate reference is given – Zwinger et al. (2007). Also, the history of modifications of the rheological law is described in a little more detail in the Introduction:

Gagliardini and Meyssonnier (1997) adapted the rheological law of Duva and Crow (1994) for a cold glacier with a thick firn layer and implemented it in a 2D dynamical finite element model for Dôme du Goûter. Further, the firn rheological law of Gagliardini and Meyssonnier (1997) was applied in 2D and 3D finite element models for Colle Gnifetti glacier saddle (Monte Rosa, Swiss/Italian Alps) in the work of Lüthi and Funk (2000). Zwinger et al. (2007) modified the rheological relations of Gagliardini and Meyssonnier (1997) for 3D thermomechanically coupled Stokes flow model implemented based on Elmer/Ice and applied the model to a crater glacier at Ushkovsky Volcano.

p8 L136 You forgot to mention what is $T_0$.

$T_0$ is mentioned in Table 1.

p9 L13 and below The way you are presenting the constitutive laws/field equations is kind of strange to me. Equation (13) alone is sufficient for pure ice, as it reduces to the usual Glen's law when $\rho = \rho_i$. For pure ice, the incompressibility assumption leads to div $\mathbf{v} = 0$, which implies that the strain rate tensor is purely deviatoric. However, when dealing with compressible firn, an additional equation for the spheric parts of the stress (i.e., $p$) and strain rate (i.e., div $\mathbf{v}$) tensors is required to close the constitutive relationship. This equation is:

$$p = -\frac{1}{b}(2A)^{-\frac{1}{n}}\gamma^{\frac{1-n}{n}}\,\mathrm{div}\,\mathbf{v}. \tag{1}$$

I agree that combining this equaiton to your Eq. (11) results in your Eq. (17), but in my view, this is a constitutive law (a relationship between stresses and strain rates) rather than a field equation. I am aware that Zwinger et al. (2007) also presents your Eq. (17) as a field equation. Conversely, the missing field equation in your formulation, in my opinion, is the mass conservation equation:

$$\frac{\partial\rho}{\partial t} + \mathrm{div}\,\rho\mathbf{v} = 0. \tag{2}$$

The way the problem is presented implies that the unknowns are $(u, v, w, p)$ and the corresponding system of four scalar equations is given by the three scalar equations in your Eq. (18) together with the scalar Eq. (17). However, this formulation neglects the fact that $\rho$ is another unknown, and thus Eq. (2) is necessary to close the system. Assuming that $\rho$ is known from core measurements and can be applied uniformly across the domain is a strong assumption that should at least be mentioned and discussed (see my general comments).

We agree with your remark that it is more correct to classify this equation as a constitutive relation. Thank you for pointing this out. The corresponding correction has been made in the text. Another thing is that at the level of numerical solution, as in the works of Zwinger et al. (2007), Gagliardini and Meyssonnier (1997), this relation actually acts as a field equation (instead of div $\boldsymbol{v} = 0$ for pure ice). In our case, the fields sought are $\boldsymbol{v}$ and $p$, and in the full formulation also $T$. We discuss the absence of the mass conservation equation in our formulation of the problem in the responses to your general comments.

p9 L149 A reference to justify this parameterizaiton would be welcome.

We have added the following source reference:

Ritz, C.: Time dependent boundary conditions for calculation of temperature fields in ice sheets, in: The Physical Basis of Ice Sheet Modelling, edited by: Waddington, E. D. and Walder, J. S., IAHS Press, Wallingford, UK, 207–216, IAHS Publication No. 170, 1987.

p9 L151 Same as above. I also find it strange that the dependence of the thermal conductivity of firn on its density is not accounted for (see my general comments).

The above reference also applies here. We discuss the independence of thermal conductivity from density in our model in the response to your next comment.

p9 L161 A thermodynamic model that operates in steady state, does not account for variations in thermal conductivity with density, and is unable to handle melting and refreezing (as shown from the fact that it is expressed in terms of temperature instead of enthalpy and lacks a latent heat source/sink term) calls into question its relevance for the present study. Please refer to my general comments.

Stationarity is a general and largely unavoidable (due to insufficient data) assumption in our study. We also attempted to include the dependence of thermal conductivity on density to the model, as in the work of Zwinger et al. (2007). The absence of ice melting in the model is discussed in the response to general comments. We added the following comment to the "Discussion":

Including the dependence of heat conductivity on density in the model was tried but did not lead to adequate results. A more effective approach was found to be to compensate for this simplified parameterization by choosing appropriate values of the flow enhancement factor.

p10 L179 I don't get this boundary condition. Normally at the bed, a non-penetration condition applies and the normal velocity is forced to zero.

Indeed, an explanation is needed here. A comment has been added in the text after this boundary condition (no. 21):

Under the BC (21), ice/firn particles are modeled as moving from the glacier surface to the bedrock in finite time, which ensures convergence in the numerical solution of the dating problem (1)–(2). Also, such a small deviation of the basal velocity from zero apparently does not affect the dating results of the overlying ice/firn (except for a thin bottom layer), as indicated by the coincidence of the age fields obtained with the $v_{\mathrm{b}}$ increased and decreased by several orders of magnitude compared to the selected value.

p11 L205-211 I am not sure this explanation desserves a full paragraph and I think that the last sentence could be removed.

      Agree. We have excluded this section and moved the information from it to the section "Model calibration". The last sentence has been removed.

p11 L218-220 The description of the numerical implementation is a bit unclear. From my understanding, the mechanical and thermodynamic problems are solved sequentially using a first-operator splitting approach until convergence is achieved (referred to as the 'steady state iterations' in Elmer). Independently of this coupling, the Stokes equation (Porous Solver in Elmer) is non-linear due to $n = 3$ in the constitutive law and needs to be linearized. The same approach applies to the heat equation, which is non-linear due to the $T$-dependence of heat capacity and thermal conductivity. In each non-linear iteration, a linear system is obtained, which can be solved using either direct or iterative methods.

      We have tried to clarify this description. Text revised:

      Differential field equations are solved numerically via their transformation to a discretized variational form (Gagliardini et al., 2013). **In the numerical implementation, the constitutive relation (11) is interpreted as a field equation with unknowns $v$ and $p$.**

          **For the thermomechanically coupled model the Stokes equation (17) (together with the Eq. (11)) and the heat transfer equation (18) are solved sequentially until convergence is achieved.** On each step of this nonlinear iteration a system of linear algebraic equations arises and needs to be solved […].

p12 L226 "a precalculated velocity field" → does that refer to the velocity field calculated by the mechanical model (I guess it does) ? Please, clarify.

      Yes, it does. We changed the sentence to:

      The dating equation (1) is solved in a final step based on **the velocity field calculated by the mechanical model.**

p13 L248 The last two samples do not appear in Table 2. Why ?

      We see no point in comparing the dating of these samples with the modeling results because of the large discrepancies. The unsuitability of the model for dating bottom ice is noted in the Discussion section. We also provide the revised text:

      Table 2 compares the dates obtained from our simulations with the 2009 ice core dating for reference horizons and the two upper basal ice samples. **The two lower ice samples are not included in Table 2 because the model significantly overestimates their ages.**

p16 L282-283 This is a bit of an overstatement. Your modelled temperature profile is actually quite far from the measured one.

      Text revised:

      The vertical temperature distribution, simulated using the thermomechanically coupled model, **is almost linear and overestimates temperatures (Fig. 5b). Thus, the mechanical coupling is weakly manifested.**

**References**

Brondex, J., Gagliardini, O., Gillet-Chaulet, F., and Chekki, M.: Comparing the long-term fate of a snow cave and a rigid container buried at Dome C, Antarctica, Cold Regions Science and Technology, 180, 103164, https://doi.org/ 10.1016/j.coldregions.2020.103164, 2020.

Gagliardini, O. and Meyssonnier, J.: Flow simulation of a firn-covered cold glacier, Annals of

Glaciology, 24, 242248, https://doi.org/10.1017/S0260305500012246, 1997.

Gilbert, A., Gagliardini, O., Vincent, C., and Wagnon, P.: A 3-D thermal regime model suitable for cold accumulation zones of polythermal mountain glaciers, Journal of Geophysical Research (Earth Surface), 119, 18761893, https://doi.org/10.1002/2014JF003199, 2014.

Licciulli, C., Bohleber, P., Lier, J., Gagliardini, O., Hoelzle, M., and Eisen, O.: A full Stokes ice-flow model to assist the interpretation of millennial-scale ice cores at the high-Alpine drilling site Colle Gnifetti, Swiss/Italian Alps, Journal of Glaciology, pp. 114, https://doi.org/10.1017/jog.2019.82, 2019.

Lüthi, M. and Funk, M.: Dating of ice cores from a high Alpine glacier with a flow model for cold firn, Annals of Glaciology, 31, 6979, https://doi.org/10.3189/172756400781820381, 2000.

Vincent, C., Vallon, M., Pinglot, J. F., Funk, M., and Reynaud, L.: Snow accumulation and ice flow at Dôme du Goûter (4300 m), Mont Blanc, French Alps, Journal of Glaciology, 43, 513521, https://doi.org/10.3189/ S0022143000035127, 1997.

Zwinger, T., Greve, R., Gagliardini, O., Shiraiwa, T., and Lyly, M.: A full Stokes-flow thermo-mechanical model for firn and ice applied to the Gorshkov crater glacier, Kamchatka, Annals of Glaciology, 45, 2937, https://doi.org/ 10.3189/172756407782282543, 2007.

**References**

Duva, J. M. and Crow, P. D.: Analysis of consolidation of reinforced materials by power-law creep, Mech. Mater., 17, 25–32, 1994.

Kutuzov, S., Lavrentiev, I., Smirnov, A., Nosenko, G., and Petrakov, D.: Volume Changes of Elbrus Glaciers From 1997 to 2017, Front. Earth Sci., 7, https://doi.org/10.3389/feart.2019.00153, 2019b.

Mikhalenko, V., Sokratov, S., Kutuzov, S., Ginot, P., Legrand, M., Preunkert, S., Lavrentiev, I., Kozachek, A., Ekaykin, A., Faïn, X., Lim, S., Schotterer, U., Lipenkov, V., and Toropov, P.: Investigation of a deep ice core from the Elbrus western plateau, the Caucasus, Russia, Cryosph., 9, 2253–2270, https://doi.org/10.5194/tc-9-2253-2015, 2015.

Mikhalenko, V. N. (Ed.): Elbrus Glaciers and Climate, Nestor-Historia Publications, Moscow-St. Petersburg, Russia, ISBN 978-5-4469-1671-9, 2020.

Salamatin, A. N., Murav'yev, Y. D., Shiraiwa, T., and Matsuoka, K.: Modelling dynamics of glaciers in volcanic craters, J. Glaciol., 46, 177–187, 2000.

Salamatin, A. N., Shiraiwa, T., Muravyev, Y. D., Kameda, T., Silantiyeva, E., and Ziganshin, M.: Dynamics and borehole temperature memory of Gorshkov Ice Cap on the summit of Ushkovsky Volcano, Kamchatka Peninsula, in: Proceedings of the International Symposium on the Atmosphere–Ocean–Cryosphere Interaction in the Sea of Okhotsk and the Surrounding Environments held at Institute of Low Temperature Science, Hokkaido University, Sapporo, Japan, 12–15 December 2000, 120–121, 2001.

Shiraiwa, T., Muravyev, Y. D., Kameda, T., Nishio, F., Toyama, Y., Takahashi, A., Ovsyannikov, A. A., Salamatin, A. N., and Yamagata, K.: Characteristics of a crater glacier at Ushkovsky volcano, Kamchatka, Russia, as revealed by the physical properties of ice cores and borehole thermometry, J. Glaciol., 47, 423–432, https://doi.org/10.3189/172756501781832061, 2001.

Stiévenard, M., Nikolaëv, V., Bol'shiyanov, D. Y., Fléhoc, C., Jouzel, J., Klementyev, O. L., and Souchez, R.: Pleistocene ice at the bottom of the Vavilov ice cap, Severnaya Zemlya, Russian Arctic, J. Glaciol., 42 (142), 403–406, https://doi.org/10.3189/S0022143000003385, 1996.

---

## Author Comment (AC2)

**Reply on Referee Comment 2**

Dear Dr. Martin Lüthi,

We are extremely thankful to you for such a great effort in reviewing our manuscript, as well as for many constructive comments that served to improve our work, and for useful ideas for future research! We are also very grateful for the overall positive assessment of our work! We have tried to take into account the proposed revision as fully as possible and have prepared a new version of the manuscript. Below, the text of your review is highlighted in blue, our responses are in black, and the corrected or added fragments of the manuscript text are in purple.

Dear collegues

This is an interesting and important analysis of the flow behavior of a firn-covered large glacier, where an ice core for climatic analysis has been drilled. The paper is nicely written and comprehensive. The discussion needs some more consideration of the shortcomings of neglecting bubble close-off and a static climate.

My recommendation is to publish the manuscript after taking into account the comments below.

**General comments**

- The abstract is quite long and should be considerably shortened. It contains repetitions and too many details.

We have shortened the abstract. The revised text is below:

The glaciers of Mount Elbrus (Caucasus) contain paleoclimatic and paleoenvironmental information representative of a vast region. Negligible seasonal melting in the near-summit area of Elbrus ensures excellent preservation of climatic signals. In 2009, a 182.65 meter long ice core was obtained from the glacier on the near-summit Western Plateau (WP) of Elbrus. The upper part and basal samples of the core were dated. In this work, a three-dimensional (3D) steady state thermomechanically coupled Stokes flow model for a cold glacier with a rheological law accounting for firn densification, calibrated based on the ice core dating, was applied to model the velocity field and the corresponding distribution of the age of the ice in the central part of the WP. We performed multiple model runs, varying boundary conditions (BCs), ice viscosity, and the inclusion of thermomechanical coupling. The Elmer/Ice software was used for numerical simulation. The model quite accurately reproduces the age of the ice according to ice core data to a depth of 150 m (up to 170 years). Below, the age of the ice increases sharply and the discrepancies in dating between different modeling scenarios become larger. Overall, the simulated ages fell within 68.2 % confidence intervals for the ages of near-bottom ice samples (mean radiocarbon age 1–2 ka). The model is not applicable for dating the lowermost ice layer (3–4 m thick). Future model improvements should focus on accounting for potential melting and identifying areas containing the oldest ice.

- "full Stokes" does not exist (but seems to be marketing jargon of Elmer Ice) -> these are just the Stokes equation from fluid mechanics. There exist "reduced" equations, omitting some terms due to scaling arguments.

Yes, it is possible to omit the word "full". By the way, the term "full Stokes flow problem" is used, for example, by Greve and Blatter (2009) even outside the context of Elmer/Ice. Also, we have taken into account all the comments below on this topic.

- "ice age" usually means a geological epoch. Better replace this term everywhere with "the age of the ice"

Agree. Corrections have been made everywhere.

- citation style should be adapted: Often \citet or \citep!

Done.

- Section 3 is out of place, and is also partly repeated in Section 4. Maybe convert to a table, or relate it better to the rest of the manuscript.

In our opinion, Section 3 is a logical continuation of the description of the study area and field data. In addition, the information presented in it is necessary for understanding the operation of the model. We appreciate the idea of converting it to a table, but Section 3 probably contains too much verbal information to be presented in tabular form. Therefore, we would prefer to keep Section 3 as it is. Also, to avoid repetitions and to clarify the wording, we have changed the beginning of Section 4. The revised text is presented here:

Ice/firn flow modeling with subsequent dating was performed in a 3D domain (Fig. 2). The domain is limited by that part of the glacier on the WP, for which DEMs of both the surface and the bed are available. The computational domain is bounded by three surfaces: a part of the glacier surface; the lateral surface of the domain (the vertical "wall"); a part of the glacier base.

- also in section 4 and 5 information is not given in a logcial order. You should rater describe, in this order:

- geometry, density, temperature etc

- numerical approach & solver etc

- discretization

- boundary conditions

Partly agree. We think that it would be most logical to first fully present the analytical formulation of the problem and then move on to the numerical implementation of the model. Therefore, we would prefer that the subsection "Boundary conditions" precedes the numerical approach. Also, in our opinion, discretization logically precedes the methods for solving linear systems, etc. At the same time, we agree that the existing sequence of presentation is not optimal. We have moved the information on discretization from Section 4 "Spatial structure" to Section 6 "Numerical methods".

To obtain a reasoable age of the ice at the base, the air bubble pressure after close-off has to be taken into account. This was pioneered by Pimienta, and implemented in e.g. Lüthi & Funk (2000). Without that effect, the ice at the botton cannot be dated correctly, and the age is much too old. Also, varying basal melt could easily be used to control the age of the ice at the bottom. If it is too old in the model, just slightly increase the melt.

Overall, this is a very nice and conclusive study on the important topic of dating a climatologically relevant ice core. The study has a few shortcomings, especially wrt. the model. These problems are mostly discussed, and are largely due to the lack of data to constrain the model. These include flow velocities and climate data to constrain the long-term thermal and dynamical evolution of the glacier. I think that at this stage it is not useful or necessary to implement bubble close-off in the Elmer Ice code base (but this needs urgently be done for subsequent studies).

The discussion should not only list, but also clearly work out the effects of neglecting in the model the effects of bubble close-off, and of assuming a high and constant basal heat flow, as well as assuming steady climate conditions.

Agree. We discuss these issues in the comments below.

**Specific comments**

22 If the area is so large and encompasses North Africa and Asia, it is not representative for any of those.

We agree that the wording should be clarified. It was meant that this paleoarchive for each of these regions contains some proxy of atmospheric phenomena (precipitation, dust, etc.). We corrected the sentence as follows:

The glaciers of Mt. Elbrus offer a unique **paleoclimate archive that traces signals from a large region,** including the North Caucasus, the Black Sea region, Southeastern Europe, North Africa, and the Middle East (Mikhalenko et al., 2024; Kutuzov et al., 2019a).

34 does the part after "while" refer to measurements? Or the model?

This part refers to the model results. Here is the corrected phrase:

According to an estimate based on a two-dimensional (2D) analytical model of Salamatin et al. (2000), the age of the ice at the bottom of the drilling site does not exceed 350–400 years and the age of the basal ice in the deepest part of the glacier (more than 250 m deep) is about 660 years (Mikhalenko et al., 2015).

39/45/54 and other places "full Stokes" -> Stokes

Done.

40 "the accumulation record"

Corrected.

49 "Mt. Dôme du Goûter" -> leave away Mt.

Corrected.

50 " Blank," -> Blanc

Corrected.

61 the glaciated area

Corrected.

62 a stray parenthesis

Corrected.

74 Better put the information on Pleiades in the Acknowledgements (does not fit in the main text)

Agree. We have moved the sentence with information about providing the Pléiades DEM to the Acknowledgments.

79 Here, I was expecting the Pleiades DEM for the surface
In fact, this includes the Pléiades DEM for the surface. We have corrected this data list item as follows:
1. DEMs of the glacier surface and bedrock with a cell size of 10 × 10 m, both obtained in 2017.

79 How do you get a 10x10 m bedrock from a radar survey? Was this somehow gridded?
Text revised:
The results of a series of ground-based radar surveys at a frequency of 20 MHz in 2005, 2007, and 2017 show a significant ice thickness and a crater shape of the underlying bedrock. The maximum depth is 255±8 m at the central part of the plateau, with minimum values of about 60 m near the edge. **The GPR survey used in this study was conducted in July 2017. The ice thickness map was completed using empirical Bayesian kriging interpolation (Kutuzov et al., 2019b).**

Figure 1, caption: Coordinates are NOT given in UTM, but this would be much more useful!
Corrected.

89 So, this is a triangulation of the domain. How was this done, with Triangle? You should also mention that you need this grid for the FE-model.
Text revised:
**For finite element calculations,** a flat computational grid (footprint) was created **using the mesh generator Gmsh** within the contour (shown in Fig. 1) of the area of the WP covered with DEMs.

92 Does this mean that you have prismatic elements, which are problematic for the incompressible flow due to LBB stability criterion?
Yes, we use prismatic finite elements. In our compressible formulation, convergence is achieved in all cases.

Also, the mesh resolution with depth is wildly varying. I can see the advantage of extruding a mesh in the vertical, but a TET4 mesh would likely be much more suited for the computations at hand.
Thank you for the idea for our future investigations! Indeed, the mesh resolution is strongly depth-dependent, and we have added the following information in the text:
The typical node spacing at the refined boundary is of order 0.1 m (less than 1 m), and at the coarser boundary it is of order 1 m (less than 10 m).

95 This argument is not clear. It would be trivial to shift the bedrock up by 5 m, and then retain the exact horizontal position of the borehole.
We are grateful to the reviewer for this suggestion! This is a very straightforward and correct idea. Our only doubt is that this approach is likely to underestimate (even if only slightly) the ice thickness in the model (assuming that the radar data are not subject to a systematic error towards overestimation of the thickness). Our experience of numerical simulations with a similar model, but other objects, shows that varying the ice thickness with other parameters unchanged has a very significant effect on the velocity values. Unfortunately, implementing this approach would

require performing the full simulation from the very beginning. However, we will certainly apply this technique in further research.

This is crucial, since mainly surface slope drives ice flow, especially in delicate saddle-like topographies like the one modeled in this study. Point a is on a clear slope, while point b is not.

As for the simulations already performed, to compensate for possible errors caused by shifting the point of modeling in combination with heterogeneity of the subglacial relief, we have a calibration parameter (flow enhancement factor, discussed below), as well as the possibility to vary the lateral boundary condition and the type of model (purely mechanical/thermodynamically coupled).

Also, radio echo data was likely not correctly interpreted (vertical instead of perpendicular to the surface), and depends a lot on the velocity model assumed, which depends on the density structure & temperature.

Sorry, but your point is not clear enough for us. The methodology for processing GPR data and error estimation is discussed in the work of Kutuzov et al. (2019b).

99, Tab 1 "Mathematical model"  better: "Numerical model" or "Ice flow model"

We agree with the remark. We have replaced "mathematical model" with "ice flow model" in the title and the text of the section. Below is the corrected text:

Under the assumption of steady state, the velocity distribution in the glacier allows one to calculate the time required for each ice/firn particle to move from the glacier surface to its current position. The calculation of the velocity field was performed on the basis of a 3D stationary Stokes model with the rheological law of Gagliardini and Meyssonnier (1997) for a compressible nonlinear viscous medium (ice/firn). We have applied both purely mechanical (isothermal) and thermomechanically coupled ice flow models. Information on the quantities used in the models is given in Table 1.

Tab 1: there is way too much information in this table at the same time it is totally unclear what type of relations were used.

We believe that it should be convenient for readers to have all the numerous parameters of the model in a single table, which can be used as a reference. We also believe that indicating the functional dependencies in the table even without giving their explicit form (due to their cumbersomeness) should make it easier to understand the model. Of course, explicit relationships between the quantities of the model can be determined by analyzing the formulas in the text. Therefore, we would prefer to keep the table in its current form.

113 This "full Stokes..." appears here for the 3rd time. Rather describe it properly at some point.

Since there are no approximations to describe in our Stokes problem setting, we added a brief remark at the beginning of the section "Ice flow model":

The model is applied in its full version, without scaling and excluding any terms in the equations below.

120 please show the data and the approximations in a figure.

The following plot has been added to the text:

[Figure]

125 Show the equations that are solved (maybe in an appendix). Not everyone wants to look up these equations. Are these the exactly same equations? Then why show them. If not: what is different?

> Yes, these are exactly the functions used by Gagliardini and Meyssonnier (1997). In general, we have chosen to write down explicitly all the formulas we use. In addition, it will really save interested readers from having to look for them in other works. Also, to be consistent in shortening the description of the model, we should exclude a number of other formulas, for example, the tensor invariant and the Arrhenius law, limiting ourselves to indicating references. We fear that this would make the presentation less clear.

130 The derivation of the tensor invariant makes no sense if it is not presented in the context of the flow relation used. Either show all, or nothing (but show in the appendix what exact equations were used).

> We agree that this information is redundant. We excluded from the test the definition and decomposition of the strain-rate tensor. As for transferring the formulas to the appendix, we would prefer not to do this, since some elements of the model (for example, the flow enhancement factor) are mentioned in the presentation of the results, and readers would have to refer to the appendix to fully understand the content of the sections of the article preceding it.

130 It is confusing using "D" for several things, and also "T".

> Partially agree. In our opinion, such notations have the advantage of being easily associated with the corresponding operations (deviator and transpose). In addition, when denoting these operations, we use different fonts than when denoting the strain-rate tensor and temperature. Therefore, we would prefer to keep the existing designation. Also, when correcting the text according to your previous comment, we excluded the only formula with transpose.

 This is wrong in the context of a compressible flow relation. "p" is a Lagrange multiplier in a purely incompressible flow law. Here a compressible law is used, were isotropic and deviatioric parts are mixed. See e.g. Gagliardini & Meyssonnier (1997), Lüthi & Funk (2000)

> Sorry, we may not understand you, but we do not see a mistake here. The implementation of the porous flow law in Elmer/Ice uses the same definition of pressure as in the incompressible case $p = -\operatorname{tr}\boldsymbol{\sigma}/3$
> (https://elmerfem.org/elmerice/wiki/lib/exe/fetch.php?media=solvers:poroussolver.pdf). Thus, $p$ represents the total pressure in the compressible fluid (Greve and Blatter, 2009). The $p$ defined in this way differs only in sign from the pressure in the work of Gagliardini and Meyssonnier (1997). Also, $p = -\sigma_{\mathrm{m}}$, where $\sigma_{\mathrm{m}}$ is the mean pressure in Lüthi and Funk (2000). In addition, this definition of pressure is also used, for example, by Zwinger at al. (2007) and Brondex et al. (2020). In the text we have added an explanation regarding pressure:
>
> After decomposing the Cauchy stress tensor into an isotropic and a deviatoric parts $\boldsymbol{\sigma} = -p\mathbf{I} + \boldsymbol{\sigma}^{\mathrm{D}}$, where $p = -\operatorname{tr}\boldsymbol{\sigma}/3$ we can write the rheological law in general form [...].

147 This is Glen's flow law for INCOMPRESSIBLE ice. You should write down the proper Duva-Crow flow law!

> Again, we may not have understood your point, but this relation is valid for both incompressible (for which $\mathbf{D}^{\mathrm{D}} = \mathbf{D}$) and compressible fluids. As for the Duva–Crow flow law, its analogue is obtained by substituting the formula for viscosity (and then other expressions) into this general flow law. It seemed to us that it is better to introduce the relations sequentially than to combine them.

150 References for the parametrizations of c and \kappa should be given. Is T in K or degC?

> We have added the following source reference:
>
> Ritz, C.: Time dependent boundary conditions for calculation of temperature fields in ice sheets, in: The Physical Basis of Ice Sheet Modelling, edited by: Waddington, E. D. and Walder, J. S., IAHS Press, Wallingford, UK, 207–216, IAHS Publication No. 170, 1987.
>
> T is in K, as given in the Units column of Table 1.

Also, these values can be deleted in Tab 1. It is important to notice that both thermal quantities are strongly dependent on density.

> We would prefer that Table 1 contained information on all parameters used. We also attempted to include the dependence of thermal conductivity on density to the model, as in the work of Zwinger et al. (2007). Added the following description to "Discussion":
>
> Including the dependence of heat conductivity on density in the model was tried but did not lead to adequate results. A more effective approach was found to be to compensate for this simplified parameterization by choosing appropriate values of the flow enhancement factor.

154 This equation is wrong, it is k, not \kappa (diffusivity) Sorry, I see that \kappa is used for conductivity here, but it is convention to use \kappa for diffusivity and k for conductivity.

> Regarding the notations, we mainly followed the work of Greve and Blatter (2009). The heat conductivity is denoted there by 'kappa'. In general, we are used to seeing both designations for the heat conductivity. We replaced 'kappa' with '$k$' throughout the text.

156 these are the conservation of mass equation (the volume is changing!)

We agree that the term we used is inappropriate and have excluded it. Also, as recommended by Reviewer 1, we have moved this equation to the "Constitutive relations" subsection.

**165 Are these b.c. varying with depth? Already mention it here, I see Eq (26).**

Indeed, the BC (25)–(26) is varying with depth in some model cases. In the first sentence of the section (line 165) we only intended to indicate the general structure of the BCs. It seems to us that there is no need to provide any additional information about the BCs right here, especially since all the information is presented below in the same section.

**179 What is the rationale for a non-zero vertical velocity, but a zero horizontal velocity?**

Indeed, an explanation is needed here. We set the normal velocity on the bed to be non-zero, and the tangential velocity to be zero. A comment has been added in the text after this boundary condition (no. 22):

Under the BC (22), ice/firn particles are modeled as moving from the glacier surface to the bedrock in finite time, which ensures convergence in the numerical solution of the dating problem (1)–(2). Also, such a small deviation of the basal velocity from zero apparently does not affect the dating results of the overlying ice/firn (except for a thin bottom layer), as indicated by the coincidence of the age fields obtained with the $v_\mathrm{b}$ increased and decreased by several orders of magnitude compared to the selected value.

**198 This shape of the velocity profile is only reasonable below the firn-ice transition, as can be seen e.g. in Fig 4a.**

We agree that further refinement of this boundary condition makes sense in the future. However, in the absence of velocity data on the glacier, this does not seem to be so significant.

**207 This is not a reduced flow model, you solve the full flow model for a time-constant T-distribution.**

We agree that such an interpretation is possible. We have excluded the section "Reduced model" (as recommended by Reviewer 1) and moved the information from it to the section "Model calibration". Here is the revised text:

All the simulation variants produced on the basis of the thermomechanically coupled model were also repeated by means of a purely mechanical model, i.e. a flow model with the heat transfer block switched off.

The simplification of the complete model is as follows. The ice/firn temperature is assumed to be constant [...].

**211 The viscosity must be dependent on the density. This is absolutely crucial. If you use Duva-Crow it is given by a(phi) and b(phi).**

The viscosity is dependent on the density in our model (see Eq. (11)).

**213 This section should be part of the Methods section, and most of the text appears here for the 4th time.**

We agree in essence, although adding the Methods section would make the structure of sections and subsections perhaps too cumbersome. We have rewritten this section and, in particular, removed repetitions. Text revised:

Differential field equations are solved numerically via their transformation to a discretized variational form (Gagliardini et al., 2013). **In the numerical implementation, the constitutive relation (11) is interpreted as a field equation with unknowns $v$ and $p$.**

**For the thermomechanically coupled model the Stokes equation (17) (together with the Eq. (11)) and the heat transfer equation (18) are solved sequentially until convergence is achieved.** On each step of this nonlinear iteration a system of linear algebraic equations arises and needs to be solved [...].

230 What is the role of the flow-enhancement factor? Is this needed at all? How is it applied, to firn and ice simultaneously? This would lead to densification values that are not compatible with the measured densities anymore.

The enhancement factor is absolutely necessary – it is the only free parameter of the model, apart from the model type, the boundary conditions and the characteristics of the numerical implementation. It is a selectable constant and applies to firn and ice simultaneously, which, however, does not lead to contradictions, since it corrects the viscosity and not the density distribution. Also, we have made the following addition to the text:

Changing the enhancement factor results in a general increase/decrease in ice/firn velocities. The velocities, in turn, affect the age field and, in particular, the configuration of the age–depth curve at the studied location. For the purely mechanical model, we varied the enhancement factor in the range from 0.15 to 0.35; for the thermomechanically coupled model, in the range from 0.01 to 0.05. Choosing the value outside these ranges resulted in a significant discrepancy between the model and the empirical age–depth dependencies.

245 Figure 4C merits some attention. This temperature profile is truly exceptional with 18 K temperature difference on only 180 m! Such high temperature gradients and heat fluxes are truly remarkable. Why are they occurring? Is the assumed basal heat flux of 340 mW/m2 really realistic? In mountains, the vertical heat flux is often much reduced due to topography.

Here is a possible explanation: *"This value is 4–5 times higher than the average heat flux density for the Earth's surface and higher than the mean value for central Caucasus, and may be associated with a heat magma chamber of the Elbrus volcano"* (Mikhalenko et al., 2015).

251 Write out "Figure" everywhere in the text, and abbreviate it in parentheses.

When providing references to figures, we relied on the following author guidelines: *The abbreviation "Fig." should be used when it appears in running text and should be followed by a number unless it comes at the beginning of a sentence, e.g.: "The results are depicted in Fig. 5. Figure 9 reveals that..."* (https://www.the-cryosphere.net/submission.html#figurestables). However, we noticed and corrected a discrepancy with the above rule in line 245. The corrected version is:

In Fig. 5b, the dating of the upper 168.6 m section [...].

255 Since we cannot see anything useful in Fig 4b, please indicate what deviation the age of the basal ice has. Is it too old?

Yes, it is too old. Now we have mentioned this in the text:

The age of the basal ice is formally equal to 10,000 years due to the upper limit specified in the numerical solution.

280 wrong parens

Corrected.

280 But they nicely agree for other sites, such as Colle Gnifetti and Col du Dome, with similar model setups (Lüthi, Liciulli, Gagliardini, …). The problem with the deepest layers is that they might be remnants from a time when the geometry of the glacier was very different, the ice divides were shifted and the flow regime was altered.

Agree. The text will be supplemented with a review and discussion of the effect of bubble close-off.

Also, as mentioned above, neglecting the effect of bubble close-off (not yet implemented in Elmer-Ice?) strongly affects the age at depth.

Unfortunately, we do not have the data to account for this effect in the dating problem. We are aware of only one attempt to implement the bubble closure effect in Elmer/Ice (Liciulli et al., 2020).

Figure 4b: There is nothing useful to see in this figure. Show the top and bottom parts separately. Additionally/alternatively, you could use a logarithimc age scale. Also indicate the reference dates (volcanoes) with dots.

Thank you for this very useful suggestion! This is done (see below).

Figure 4c: Please indicate 0 degrees, and also the pressure melting temperature at the base. From the plot one cannot determine the basal temperature.

The basal temperature according to the model turns out to be equal to the pressure melting temperature (–0.14 °C), which is now noted in the text. Since the latter value differs so little from 0 °C, it makes no sense to indicate it, in our opinion. Additionally, taking into account Reviewer 1's comment, we removed the velocity plot as it does not seem very informative in the context of our work. Below is the plot revised:

[Figure]

For a steady state model with density variation you will never get a straight line, since the vertical heat flux is constant, but you need a higher gradient in firn. The measured temperature profile also looks advective, which should have been captured by the model.

In fact, this profile is slightly nonlinear, it's just that its nonlinearity is very small. Indeed, thermomechanical coupling has little effect in our simulation. In general, for us it was just one of a series of numerical experiments. In our experience, in some cases, using a purely mechanical model to estimate the age of ice yielded better results in terms of matching ice core chronology, compared with the case of thermomechanical coupling. Namely, when scaling the rheological function in a coupled model, large corrections were required.

Finally, the spatially constant vertical heat flux b.c. in the model is not realistic in a mountain topography (e.g. Lüthi, 2001).

Since we have the heat flux value at only one point, the value of the vertical gradient of the heat flux would remain hypothetical. We will take this into account in further studies and consider it in more detail in the "Discussion".

Figure 4 caption: what is this stray "vertical a"??

We have corrected the caption to:

**Figure 5.** Modeled age (a), and temperature (b) **vertical profiles for the location *a*** (see Fig. 4) and empirical data.

Also, the number of this figure has changed.

**Citation**: https://doi.org/10.5194/egusphere-2024-3955-RC2

**References**

Brondex, J., Gagliardini, O., Gillet-Chaulet, F., and Chekki, M.: Comparing the long-term fate of a snow cave and a rigid container buried at Dome C, Antarctica, Cold Regions Science and Technology, 180, 103164, https://doi.org/ 10.1016/j.coldregions.2020.103164, 2020.

Gagliardini, O. and Meyssonnier, J.: Flow simulation of a firn-covered cold glacier, Ann. Glaciol., 24, 242–248, https://doi.org/10.3189/S0260305500012246, 1997.

Greve, R. and Blatter, H.: Dynamics of Ice Sheets and Glaciers, Springer, Berlin, Germany, 2009.

Kutuzov, S., Legrand, M., Preunkert, S., Ginot, P., Mikhalenko, V., Shukurov, K., Poliukhov, A., and Toropov, P.: The Elbrus (Caucasus, Russia) ice core record-Part 2: History of desert dust deposition, Atmos. Chem. Phys., 19, 14133–14148, https://doi.org/10.5194/acp-19-14133-2019, 2019a.

Kutuzov, S., Lavrentiev, I., Smirnov, A., Nosenko, G., and Petrakov, D.: Volume Changes of Elbrus Glaciers From 1997 to 2017, Front. Earth Sci., 7, https://doi.org/10.3389/feart.2019.00153, 2019b.

Licciulli, C., Bohleber, P., Lier, J., Gagliardini, O., Hoelzle, M., and Eisen, O.: A full Stokes ice-flow model to assist the interpretation of millennial-scale ice cores at the high-Alpine drilling site Colle Gnifetti, Swiss/Italian Alps, J. Glaciol., 66, 35--48, https://doi.org/10.1017/jog.2019.82, 2020.

Lüthi, M. and Funk, M.: Dating of ice cores from a high Alpine glacier with a flow model for cold firn, Ann. Glaciol., 31, 69--79, https://doi.org/10. 3189/172756400781820381, 2000.

Mikhalenko, V., Sokratov, S., Kutuzov, S., Ginot, P., Legrand, M., Preunkert, S., Lavrentiev, I., Kozachek, A., Ekaykin, A., Faïn, X., Lim, S., Schotterer, U., Lipenkov, V., and Toropov, P.: Investigation of a deep ice core from the Elbrus western plateau, the Caucasus, Russia, Cryosph., 9, 2253–2270, https://doi.org/10.5194/tc-9-2253-2015, 2015.

Mikhalenko, V., Kutuzov, S., Toropov, P., Legrand, M., Sokratov, S., Chernyakov, G., Lavrentiev, I., Preunkert, S., Kozachek, A., Vorobiev, M., Khairedinova, A., and Lipenkov, V.: Accumulation rates over the past 260 years archived in Elbrus ice core, Caucasus, Clim. Past, 20, 237–255, https://doi.org/10.5194/cp-20-237-2024, 2024.

Salamatin, A. N., Murav'yev, Y. D., Shiraiwa, T., and Matsuoka, K.: Modelling dynamics of glaciers in volcanic craters, J. Glaciol., 46, 177–187, 2000.

Zwinger, T., Greve, R., Gagliardini, O., Shiraiwa, T., and Lyly, M.: A full Stokes-flow thermo-mechanical model for firn and ice applied to the Gorshkov crater glacier, Kamchatka, Ann. Glaciol., 45, 29--37, https://doi.org/10.3189/172756407782282543, 2007.

---

## Author Response (AR1)

Dear Editor,

We agree that the work by Gilbert et al. (2014) is absolutely relevant and have included a reference to it in our review of studies on the topic. As for the work by Brondex et al. (2020), it is not related to the topic of ice/firn dating and we mention it in our responses to reviewers just because it is one of the examples of the use of the same rheological law as in our study. In our opinion, there is no necessity to cite this work in the text of our manuscript. Thank you for your suggestions!

Sincerely,

Gleb Chernyakov

---

## Author Response (AR2)

**Reply on Report #1**

Dear Dr. Martin Lüthi,

We are very grateful to you for the second review of our manuscript, your positive assessment of the revisions done, and recommendation for publication after final revisions. We consider your comments on the revised version of the manuscript to be very significant and have taken them into account. Below, we provide the text of your review, our responses, and the corrected or added fragments of the manuscript.

Sincerely,
Gleb Chernyakov, on behalf of all the authors

Dear colleagues

This is the second time I read the manuscript. This version is a well-done improvement on the first version of the paper. My recommendation is to publish the article once the small comments below have been taken into account.

Sincerely,
Martin Lüthi

General comments

What is the rationale for using Point "a" for the distributions, but "b" for the data? Shouldn't be all evaluated at "b"? At lines 95ff some of this rationale is given, but it should be more prominently advertised, and be re-formulated such that it is clear from the beginning, why these positions are different. Just start the paragraph starting on line 93 with: "We chose to evaluate the model results at position "a" instead of position "b", where the bore hole was located, for the following reasons."

*Thank you for the suggestion! To emphasize from the outset that the change in position is due to a discrepancy between different data, we have supplemented the paragraph with the introductory phrase you suggested:*

We chose to evaluate the model results at position *a* instead of position *b*, where the borehole was located (see Fig. 2), for the following reason. According to the DEMs of the surface and bedrock, obtained by interpolation of the GPR data profiles the ice thickness at the 2009 drilling site location is 6.1 m greater than the ice core length. This difference is within the accuracy of the GPR data but creates the discrepancy in modeled ice thickness.

In Equation (2) you describe a purely advective problem. How was this solved, they very often lead to numerical instability. Why did you not use particle tracking from the borehole back to the surface (as was done in e.g. Lüthi (2000)).

> *Equation (1) with boundary condition (2) is meant here. We discuss the approach to solving this problem in the last paragraph of the Numerical methods section: "The equation was discretized via discontinuous Galerkin method and the resulting linear algebraic systems were solved by a direct method for the thermomechanically coupled model and by BiCGStab with ILU(1) preconditioner for the purely mechanical model". The computations converged and numerical instability was not observed. Regarding the backward trajectories, there was no necessity to explicitly track them for modeling the age field, since we applied the well-functioning advection– reaction Elmer solver. However, we previously performed backward trajectories simulation for this borehole in order to account for the upstream effect in the reconstruction of the accumulation rates at the Elbrus Western Plateau (Mikhalenko et al., 2024).*

The use of an enhancement factor E is somewhat unsatisfactory. This very large discrepancy from literature values should be discussed in detail. It appears, that the outflow boundary condition has a very strong influence on the resulting dating (Figure 4 clearly shows that). (Similar to what we found for Colle Gnifetti, Lüthi (2000)).

> *We agree that the issue is insufficiently clarified. The text was revised.*
>
> *For your knowledge, we came across this problem on different mountain glaciers when ice/firn fluidity is greatly overestimated in the uncorrected case (E = 1), and a strong reducing coefficient is need. To investigate the reasons for this we varied the geometric and physical parameters of the modeled domain (using another mountain glacier, but a similar model). Our modeling experiments showed that the average flow velocity is strongly dependent on glacier geometry (thickness, slope, domain size), but that density (replacing firn with pure ice), bedrock topography, and the presence/absence of lateral outflow have only a minor effect (also, Figure 4 in the manuscript corresponds to a case without lateral outflow). These results are still preliminary, so we would prefer to refrain from speculating on this topic in this article.*
>
> *We have added the following comment to the Discussion section:*
>
> The flow enhancement factor values selected as a result of model calibration turned out to be less than 1, which is not typical for glacier dynamics models (Greve and Blatter, 2009) and indicates that without appropriate correction the flow model overestimates the ice/firn fluidity. Similar results were reported previously for a crater glacier in Kamchatka where the value of the flow enhancement factor was also less than unity ($E = 1/3$) (Zwinger et al.,2007). Further analysis will be required to identify the causes of the atypical shift in the value of this parameter.

Specific comments

92 "wall"); *and * a part...

The sentence has been corrected.

347 Among all... (strange sentence, reformulate)

We have reformulated the sentence as follows:

From all the simulation results, we determined the range of modeled age–depth curves corresponding to the 2009 ice core.

**References**

Greve, R. and Blatter, H.: Dynamics of Ice Sheets and Glaciers, Springer, Berlin, Germany, 2009.

Mikhalenko, V., Kutuzov, S., Toropov, P., Legrand, M., Sokratov, S., Chernyakov, G., Lavrentiev, I., Preunkert, S., Kozachek, A., Vorobiev, M., Khairedinova, A., and Lipenkov, V.: Accumulation rates over the past 260 years archived in Elbrus ice core, Caucasus, Clim. Past, 20, 237–255, https://doi.org/10.5194/cp-20-237-2024, 2024.

Zwinger, T., Greve, R., Gagliardini, O., Shiraiwa, T., and Lyly, M.: A full Stokes-flow thermo-mechanical model for firn and ice applied to the Gorshkov crater glacier, Kamchatka, Ann. Glaciol., 45, 29--37, https://doi.org/10.3189/172756407782282543, 2007.